# MiniCORVET is a Vps8-containing early endosomal tether in Drosophila

Péter Lőrincz[1], Zsolt Lakatos[1], Ágnes Varga[1], Tamás Maruzs[2], Zsófia Simon-Vecsei[2], Zsuzsanna Darula[3], Péter Benkő[1], Gábor Csordás[2], Mónika Lippai[1], István Andó[2], Krisztina Hegedűs[1], Katalin F Medzihradszky[3], Szabolcs Takáts[1], Gábor Juhász[1,2]*

[1]Department of Anatomy, Cell and Developmental Biology, Eötvös Loránd University, Budapest, Hungary; [2]Institute of Genetics, Biological Research Centre, Hungarian Academy of Sciences, Szeged, Hungary; [3]Laboratory of Proteomics Research, Biological Research Centre, Hungarian Academy of Sciences, Szeged, Hungary

**Abstract** Yeast studies identified two heterohexameric tethering complexes, which consist of 4 shared (Vps11, Vps16, Vps18 and Vps33) and 2 specific subunits: Vps3 and Vps8 (CORVET) versus Vps39 and Vps41 (HOPS). CORVET is an early and HOPS is a late endosomal tether. The function of HOPS is well known in animal cells, while CORVET is poorly characterized. Here we show that Drosophila Vps8 is highly expressed in hemocytes and nephrocytes, and localizes to early endosomes despite the lack of a clear Vps3 homolog. We find that Vps8 forms a complex and acts together with Vps16A, Dor/Vps18 and Car/Vps33A, and loss of any of these proteins leads to fragmentation of endosomes. Surprisingly, Vps11 deletion causes enlargement of endosomes, similar to loss of the HOPS-specific subunits Vps39 and Lt/Vps41. We thus identify a 4 subunit-containing miniCORVET complex as an unconventional early endosomal tether in Drosophila.

*For correspondence: szmrt@elte.hu

**Competing interests:** The authors declare that no competing interests exist.

## Introduction

Eukaryotic cells take up extracellular material and transmembrane receptors via endocytosis. Endocytic cargo will either be recycled or degraded in lysosomes. Early (also known as sorting) endosomes represent the central hub where this decision is made. This tubulo-vesicular organ can also receive input from the Golgi or other intracellular vesicles. The fate and maturation of endosomes are regulated by several Rab GTPase proteins. Active, GTP-loaded Rabs recruit effectors to the vesicular membranes, which then facilitate the subsequent events, including vesicle targeting and homo- and heterotypic fusions. Intracellular vesicles usually acquire tethering factors and SNARE (Soluble N-ethylmaleimide sensitive factor Attachment protein REceptor) proteins that facilitate their fusion. Tethering complexes and proteins cross-link the membranes of the two vesicles/compartments, and may promote SNARE complex assembly and fusion (*Hong and Lev, 2014*; *Kümmel and Ungermann, 2014*; *Südhof and Rothman, 2009*; *Wickner and Schekman, 2008*; *Zick and Wickner, 2014*).

The tethering factors are recruited to intracellular membranes by interacting with activated small Rab GTPase (or small Rab GTPase-associated) proteins and phosphoinositides. Four endosomal tethers have been identified to date: the early endosomal tethers Rbsn-5 (Rabenosyn-5), CORVET (Class C core endosome vacuole tethering), the vertebrate-specific EEA1 (Early endosomal antigen 1), and the late endosomal tether HOPS (Homotypic vacuole fusion and protein sorting). The Rbsn-5 protein itself has a tethering function similar to EEA1, whilst CORVET and HOPS are closely related heterohexameric tethering complexes (*Kümmel and Ungermann, 2014*). These protein complexes were

first identified and characterized in yeast. CORVET binds to Vps21/Rab5 and mediates early endosomal fusions, whereas HOPS binds to Ypt7/Rab7 and mediates late endosomal, autophagosomal and vacuolar fusions (*Balderhaar et al., 2013*; *Markgraf et al., 2009*; *Peplowska et al., 2007*; *Plemel et al., 2011*; *Rieder and Emr, 1997*; *Seals et al., 2000*; *Wurmser et al., 2000*). Both share a common core of class C Vps proteins (Vps11, Vps16, Vps18 and Vps33), and the two Rab-binding subunits are located on opposite ends of these complexes (*Balderhaar and Ungermann, 2013*; *Bröcker et al., 2012*). Vps3 and Vps8 are the CORVET specific subunits that bind to Vps21/Rab5, while Vps39 and Vps41 are present in HOPS and interact with Ypt7/Rab7, respectively.

All HOPS subunits are conserved in metazoans, and the roles of this complex are very similar in mammals and Drosophila to yeast cells. Interestingly, two of the genes encoding class C proteins (Vps16 and Vps33) have been duplicated in flies and mammals, but Vps16B/VIPAR and Vps33B have been suggested to form a separate endosomal targeting complex with each other, independent of CORVET or HOPS (*van der Kant et al., 2015*; *Wartosch et al., 2015* ). Importantly, among the two CORVET-specific subunits only the homologs of Vps8 can be found in higher eukaryotes, as Vps3 seems to be a fungi-specific protein (*Klinger et al., 2013*; *Solinger and Spang, 2013*). It was recently demonstrated that one of the mammalian Vps39 homologs, Vps39-2 (also known as Tgfbrap1 or Trap1), can be found in a Vps8-containing CORVET complex (*Lachmann et al., 2014*; *Perini et al., 2014*). The function of mammalian CORVET is still poorly characterized: the only functional analysis published to date reported that siRNA knockdown of Vps8 in HeLa cells affects the fusion of only a subset of early endosomes (*Perini et al., 2014*).

Drosophila is an excellent model to study vesicular trafficking in multiple tissues of a whole animal. Others and we have shown that HOPS mediates the fusion of lysosomes with endosomes and autophagosomes, and also promotes biosynthetic trafficking to lysosomes and lysosome-related organelles such as eye pigment granules (*Akbar et al., 2009*; *Jiang et al., 2014*; *Lindmo et al., 2006*; *Pulipparacharuvil et al., 2005*; *Richardson et al., 2004*; *Sevrioukov et al., 1999*; *Takáts et al., 2014*; *Warner et al., 1998*). Several HOPS subunits were named based on the defective eye color of hypomorphic mutant animals: Vps18 is called Deep orange (Dor), Vps33A is Carnation (Car), and Vps41 is known as Light (Lt) in Drosophila (*Akbar et al., 2009*; *Lloyd et al., 1998*; *Patterson, 1932*; *Sevrioukov et al., 1999*; *Warner et al., 1998*). Regarding the CORVET-specific subunits, only a Vps8 homolog can be found in Drosophila, as no gene products show significant similarity to either yeast Vps3 or mammalian Tgfbrap1 (*Klinger et al., 2013*; *Li and Blissard, 2015*). This raises the question of how Vps8 can function: is it part of an alternative CORVET complex, or does it have a new role?

In this work, we show that Drosophila Vps8 (also known as CG10144) is highly expressed in larval hemocytes and nephrocytes, the two cell types that display very high endocytic activity to ensure continuous monitoring and filtration of the blood. Through a series of colocalization, biochemical interaction, mutant and genetic epistasis analyses, we show that Vps8 functions together with Dor/Vps18, Car/Vps33A and Vps16A to promote the fusion of early endosomes independent of Vps11 and the HOPS-specific subunits Vps39 and Lt/Vps41, therefore it is a defining subunit of a novel miniCORVET complex.

## Results

### Vps8 is expressed in larval nephrocytes and hemocytes

To analyze Vps8 function in Drosophila, we first generated a HA-tagged Vps8 reporter line whose expression is controlled by genomic Vps8 promoter sequences (*Figure 1—figure supplement 1A*). Interestingly, Vps8-HA expression was strikingly stronger in garland and pericardial nephrocytes and hemocytes than in other cell types in larvae (*Figure 1A–C*, *Figure 1—figure supplement 1B–D*). A very similar expression pattern was detected in adult tissues and pupal wings (*Figure 1—figure supplement 2*). High expression of our reporter in hemocytes relative to whole larval lysate was further confirmed by western blots (*Figure 1D*).

### Vps8 localizes to early endosomes in nephrocytes

The insect renal system includes garland and pericardial nephrocytes, which simultaneously show the characteristics of vertebrate renal glomerular podocytes and proximal tubule cells. These cells utilize

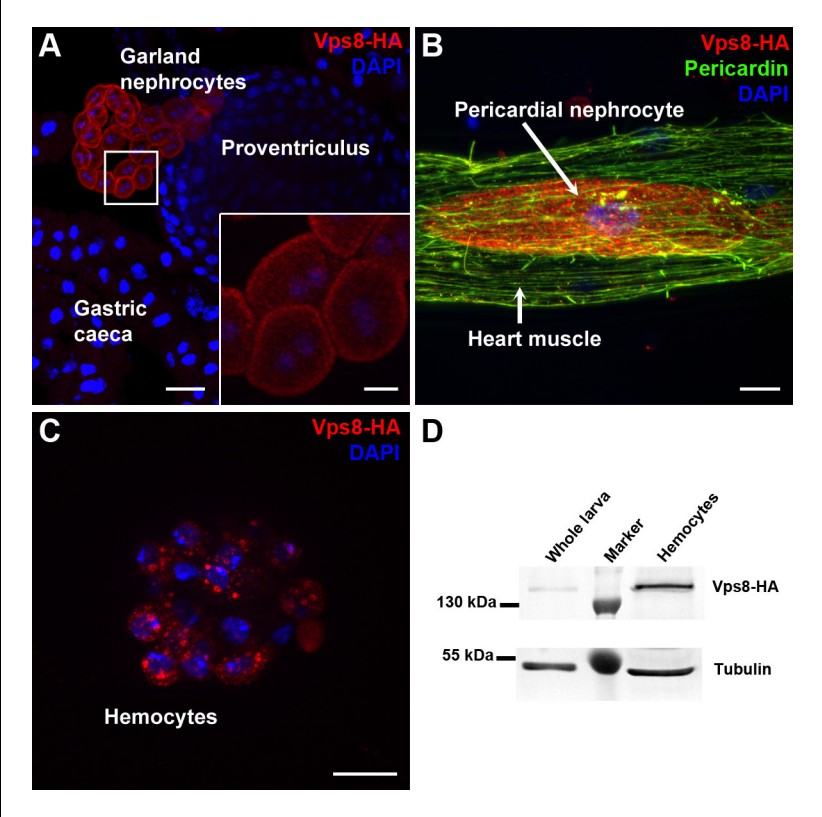

**Figure 1.** Vps8 is highly expressed in nephrocytes and hemocytes. (**A–C**) These panels show various tissues of larvae expressing a Vps8-HA reporter under the control of genomic vps8 promoter sequences. (**A**) Low magnification image of the upper gastrointestinal tract. Note the prominent expression of the reporter in garland nephrocytes. Inset shows garland cells at higher magnification. (**B**) Part of the heart tube stained with anti-Pericardin and anti-HA. The reporter is detected exclusively in pericardial nephrocytes, heart muscle cells lack reporter signal. (**C**) Vps8-HA expression in hemocytes. (**D**) Western blot showing lysates of whole larvae and isolated hemocytes demonstrates the high expression of Vps8-HA in hemocytes. Bars: (**A**): 40 μm, inset, (**B,C**): 10 μm.

The following figure supplements are available for figure 1:

**Figure supplement 1.** Vps8 is highly expressed in larval nephrocytes and hemocytes.

**Figure supplement 2.** Vps8 expression in various adult and pupal tissues.

a vertebrate-like slit diaphragm and endocytic scavenger receptors to filter the hemolymph, and store or metabolize the endocytosed material (*Weavers et al., 2009*; *Zhang et al., 2013*; *Zhuang et al., 2009*). As a consequence of this continuously high endocytic activity, the endo-lysosomal compartment fills much of the cytoplasm, and forms distinct layers in garland nephrocytes (*Figure 2A–C*). These large binucleated cells are sheathed with a basal membrane, and the underlying plasma membrane forms numerous lacunae with the slit diaphragm located at the openings. Clathrin-coated vesicles and early endosomes form under the lacunae, and are seen as the Rbsn-5 positive layer in fluorescent photographs (*Figure 2A–C*). Under this peripheral layer, large electron-lucent vesicles are found that often contain a dense core. These are also part of the endosomal system, and were traditionally referred to as α-vacuoles (*Koenig and Ikeda, 1990*; *Kosaka and Ikeda, 1983*; *Rusten et al., 2006*). Alpha-vacuoles are the largest vesicles in garland nephrocytes and are located directly under the early endosomal layer, and appear as late, Rab7-positive endosomes in fluorescent images (*Figure 2A,B*). Finally, the perinuclear β-vacuoles correspond to Lamp1-positive lysosomes (*Figure 2A,C*) (*Koenig and Ikeda, 1990*; *Kosaka and Ikeda, 1983*; *Rusten et al., 2006*).

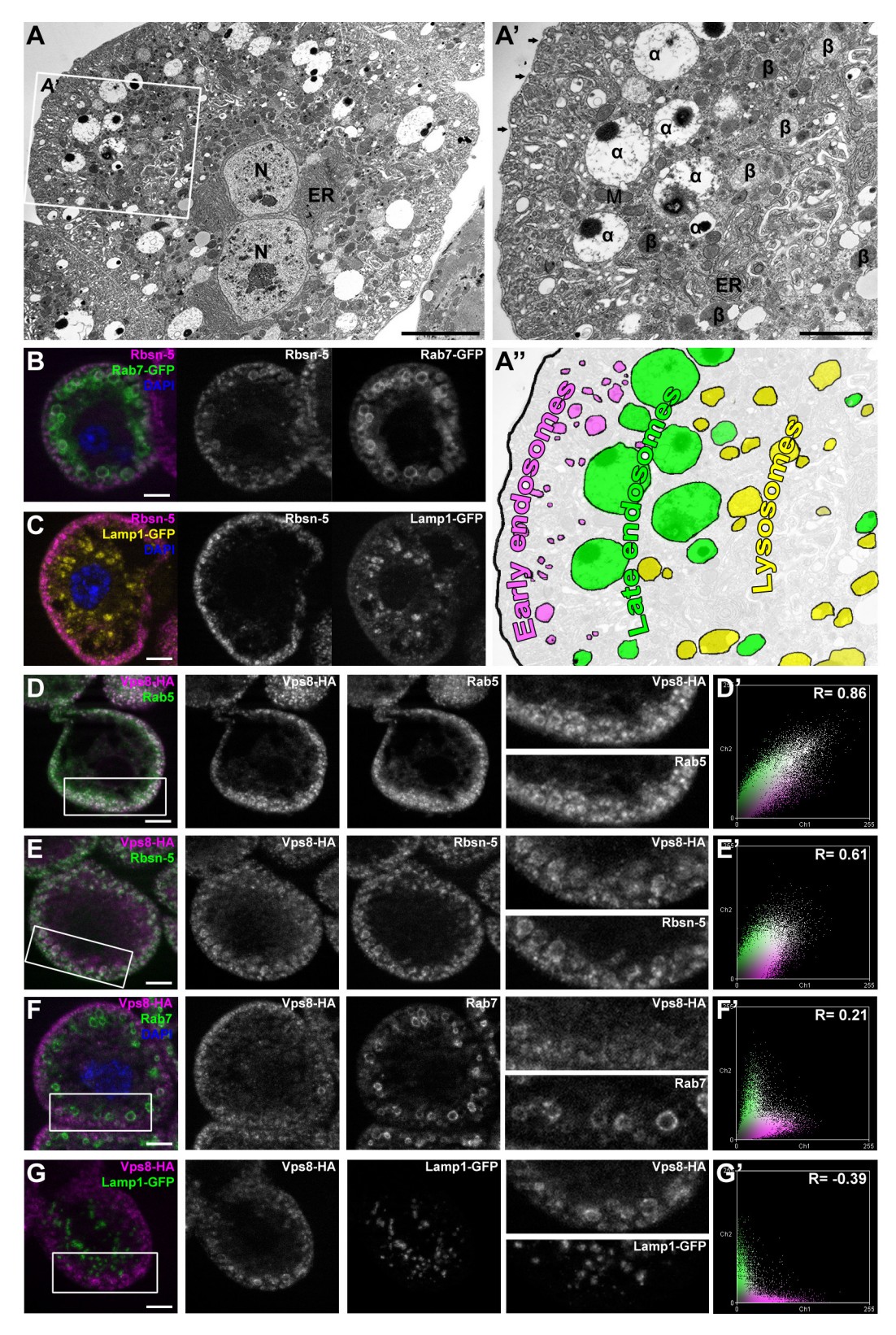

**Figure 2.** Vps8 localizes to early endosomes in nephrocytes. (A–C) These panels illustrate the general structure of garland nephrocytes. (A) Ultrastructure of a wild type garland nephrocyte. (B–C) Fluorescent images of garland cells expressing the late endosomal marker Rab7-GFP (green on

*Figure 2 continued on next page*

*Figure 2 continued*

B) or lysosomal Lamp1-GFP (yellow on C), both co-labeled with the early endosome specific anti-Rbsn-5 (magenta on both images. (A,A',A'') Garland nephrocytes are enlarged binucleate (N) cells surrounded by a basal membrane (arrows on A'), and contain numerous vesicles. Outermost the peripheral early endosomal layer is seen, which corresponds to the Rbsn-5 vesicle population in fluorescent images. Under this layer, enlarged electron-lucent vesicles are found that often contain a dense core. These are the α-vacuoles, which correspond to the Rab7-positive late endosomal layer. Beta-vacuoles are seen at the perinuclear region, and correspond to the Lamp1-positive lysosome layer. M: mitochondrion, ER: endoplasmic reticulum (D–G) Vps8-HA colocalizes with endogenous Rab5 (D) and Rbsn-5 (E), but not with Rab7 (F) or with Lamp1-GFP (G). Scatter plots show the intensity correlation profiles of Vps8-HA (magenta) with these endosomal and lysosomal markers (green) (D–G'). Pearson correlation coefficients shown at the top of these panels indicate the strong colocalization of Vps8-HA with Rab5 or Rbsn-5, only incidental colocalization with Rab7, and mutually exclusive localization with Lamp1-GFP. Bars: (A–G): 5 μm, (A'''): 2 μm.

Since the Vps8 signal had a punctate appearance in garland nephrocytes, we turned to colocalization analysis to identify the compartment where this protein may function. We found that Vps8-HA almost completely colocalizes with the early endosomal markers Rab5 and Rbsn-5 (*Figure 2D,E*), but rarely with the late endosomal Rab7 (*Figure 2F*) and never with lysosomal Lamp1 (*Figure 2G*). The occasional presence of Vps8 on smaller Rab7-positive late endosomes suggests that Vps8 may remain on endosomes until they mature into full-size late endosomes.

## Vps8 mutants are semilethal and develop melanotic tumors, unlike HOPS-specific mutants

We used Cas9/CRISPR mediated targeted mutagenesis to generate a vps8 (Flybase annotation number: CG10144) mutant. We used a double gRNA approach (both of which target the second exon of this gene, as shown in *Figure 3A*) to increase the possibility of successful mutagenesis. DNA sequencing identified a mutant line, vps8[1], which contains two small deletions, 4 and 12 base pairs, respectively, at the positions targeted by the gRNAs (*Figure 3A*). As the first deletion causes a frameshift and an immediate stop codon, only the first 39 of the 1229 amino acids of the Vps8 protein can be translated. Detailed changes in the nucleotide sequence are shown in *Supplementary file 1*. Quantitative PCR (qPCR) analysis confirmed that the frameshift mutation is present in vps8 transcripts in homozygous vps8[1]/vps8[1] mutants as well as in vps8[1]/deficiency (Df) animals, as a primer overlapping the deleted 4 nucleotides failed to amplify the wild type transcript, unlike in control and rescued animals (*Figure 3A*). Such aberrant transcripts are often degraded through nonsense-mediated decay. Indeed, qPCR using primers specific for the non-mutated region of vps8 showed that the transcript level in vps8[1]/vps8[1] mutants and in vps8[1]/Df animals decreased to about 25% compared to controls, and it was again restored in rescued animals (*Figure 3—figure supplement 1A*).

Vps8[1] mutant flies are semilethal: animals can complete metamorphosis, but most of them die as pharate adults (that is, fully formed adult flies that are unable to emerge from the pupal case). Quantification of viability revealed that 20% of homozygous vps8[1]/vps8[1] mutants and 16% of vps8[1]/Df animals eclosed, but the resulting adult flies were weak, failed to unfold their wings properly, and died within 24 hr (*Figure 3C,D*). Importantly, the eye color of vps8 mutant adult flies was similar to wild type flies (*Figure 3C,C'*), unlike hypomorphic HOPS mutants that have a pigmentation defect (*Lloyd et al., 1998*; *Patterson, 1932*; *Sevrioukov et al., 1999*; *Shestopal et al., 1997*; *Warner et al., 1998*), suggesting that Vps8 functions independent of the HOPS complex in Drosophila.

Vps8 mutant larvae had 10-fold more circulating hemocytes than control animals did, and often contained hemocyte-derived melanotic tumors in their body cavity (*Figure 3B,E,E'*). This was accompanied by the disorganization of the sessile hemocyte compartment, and the accumulation of crystal cells and lamellocytes (*Figure 3—figure supplement 1B,C*) that are only observed in large numbers during immune induction in wild type larvae (*Márkus et al., 2009*). Hematopoetic compartment-specific overexpression of PVF2, a PDGF/VEGF-like growth factor has been shown to induce hemocyte proliferation and pupal lethality in Drosophila larvae (*Munier et al., 2002*). Thus, spontaneous immune induction may contribute to the decreased viability of vps8 mutant flies.

As the semilethality, the increased blood cell number and melanotic tumor formation are all rescued by the Vps8-HA transgene, these indicate that our reporter is functional, and that the observed phenotypes are solely due to the loss of Vps8 function. Hemocyte-derived tumors are also seen in

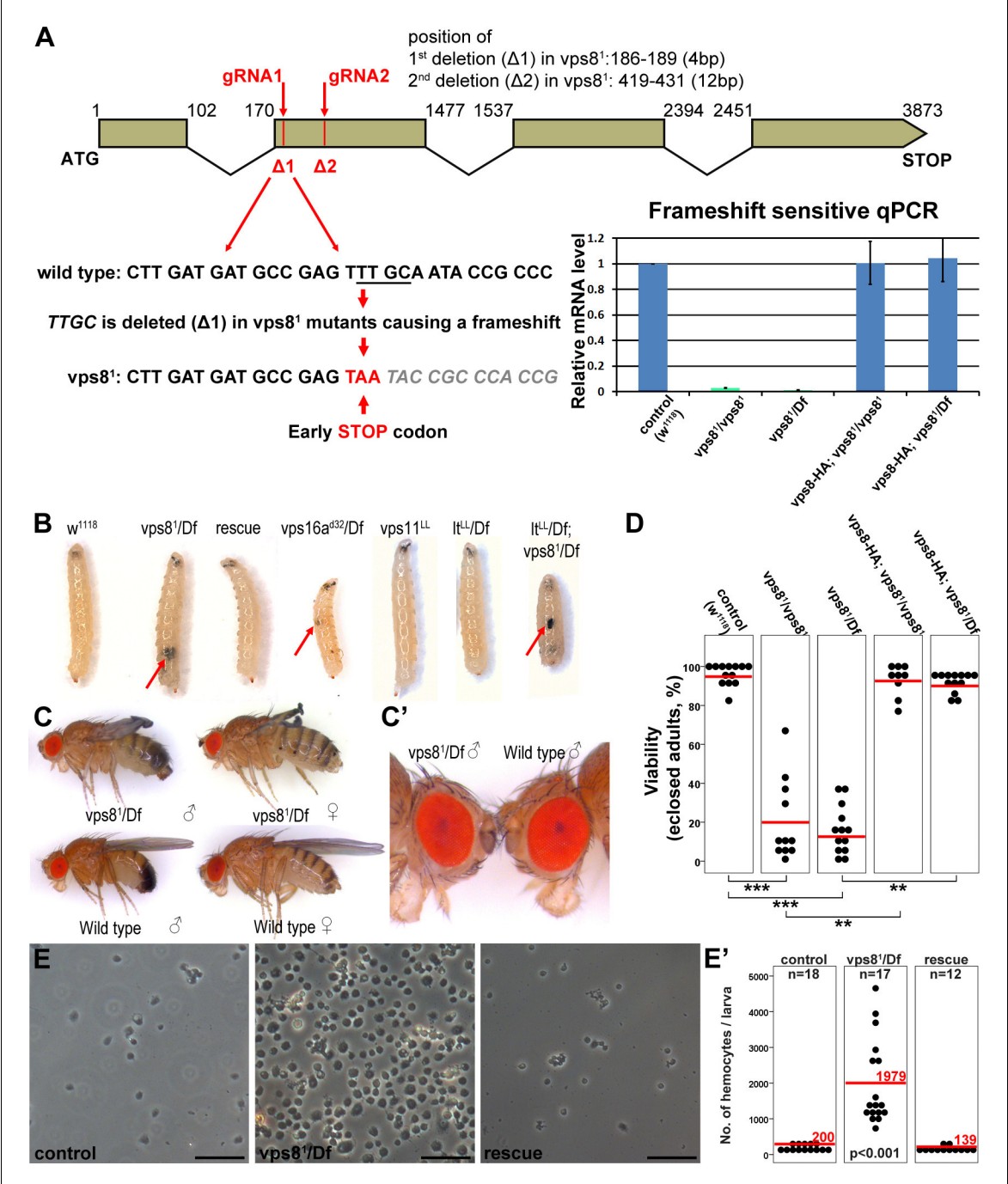

**Figure 3.** Generation of a vps8 null mutant. (**A**) Map of the vps8/cg10144 locus. CRISPR/Cas9-induced deletions in vps8[1] mutants are indicated with red lines. The sequence of the region containing the first microdeletion and the resulting early stop codon is indicated. The chart shows the results of qPCR analyses from animals of the indicated genotypes. Frameshift-sensitive primers were used to reveal that the amount of wild type vps8 transcript in mutants is negligible. The vps8 mRNA level is restored in mutants carrying the vps8-HA rescue transgene. (**B**) Photographs of larvae carrying mutations in various CORVET or HOPS subunit genes. Mutants of vps8, vps16a and double mutants lacking both vps8 and lt develop melanotic tumors (red arrows) in their body cavity. (**C,C'**) Photographs of adult vps8[1] mutant and control flies. Except for the wing spreading defects, the overall body structure of mutants is similar to wild type (**C**), including compound eye structure and pigmentation (shown enlarged in **C'**). (**D**) Dot plots show the percentage of adults that manage to eclose from the pupal case in the indicated genotypes. Each data point represents one experiment. The average number of eclosing mutant adults is decreased to 20% (vps8[1]/vps8[1]) and 16% (vps8[1]/Df), and viability is restored by the vps8-HA transgene. Red lines represent the mean, \*\*p<0.01, \*\*\*p<0.001. (**E**) Images of isolated hemocytes demonstrate that circulating hemocyte number is 10-fold higher in vps8
*Figure 3 continued on next page*

*Figure 3 continued*

mutants than in controls or mutants expressing the Vps8-HA rescue transgene. (**E'**) shows the quantification of hemocyte data, with red lines and numbers representing mean hemocyte numbers. Bars: 50 μm in (**E**).

The following figure supplement is available for figure 3:

**Figure supplement 1.** Loss of Vps8 results in ectopic immune activation.

vps16a mutants, but neither in the HOPS-specific lt mutants nor in vps11 mutants (*Figure 3B*), and melanotic tumors have also been observed for dor (*Belyaeva et al., 1982*). Importantly, double mutants of vps8 and lt are similar to vps8 mutants regarding tumor formation (*Figure 3B*). These data are compatible with the model of a yeast CORVET-like complex acting independent of HOPS in Drosophila hemocyte tumor formation, except for the phenotype of the vps11 mutant, because this gene product is part of both yeast complexes and its loss should also cause tumors similar to vps8 and vps16a mutations.

## Vps8 is required for the trafficking of endocytosed cargo in nephrocytes and hemocytes

As garland nephrocytes take up and metabolize waste products from the hemolymph, raising larvae on food complemented with silver nitrate is an easy method to analyze nephrocyte function (*Weavers et al., 2009*). Nephrocytes of larvae grown on this food normally take up silver, which is then stored in lysosomes. Lysosomal silver deposits are clearly visible as brown granules in garland cells of wild type or rescued animals (*Figure 4A,C*). In contrast, visible silver granules are almost absent from vps8[1]/Df mutant nephrocytes, indicating impaired silver uptake and/or endo-lysosomal trafficking (*Figure 4B*). To further analyze these endocytic defects, dissected nephrocytes were incubated with fluorescent dextran for 5 min. This short pulse is sufficient for the internalization and transport of dextran into large endosomes in control cells. Vps8[1]/Df cells are also able to internalize dextran, but it remains in numerous small vesicles at the periphery of the cells, pointing to a failure in endosomal trafficking (*Figure 4D–F,J*, *Figure 4—figure supplement 1E*). Similar endocytic defects were detected in garland nephrocytes of homozygous vps8[1]/vps8[1] mutant larvae in silver and dextran uptake assays, and these mutant phenotypes were again rescued by the Vps8-HA transgene (*Figure 4—figure supplement 1*).

Importantly, hemocytes of control or rescued larvae injected with fluorescently labeled Escherichia coli readily transported the bacteria into acidic, lysosomal compartments, whereas vps8[1]/Df mutants failed to acidify bacteria-containing phagosomes (*Figure 4G–I,K*). These data altogether indicate that Vps8 is required for proper endosomal trafficking in nephrocytes and blood cells.

## The endo-lysosomal compartment is fragmented in vps8 mutant nephrocytes, but not in wing disc cells

The results of the uptake assays could be explained by a failure of endosomal fusion events, which are required for the formation of large late endosomes and endolysosomes in vps8 mutant nephrocytes. In line with this, Rab7-positive late endosomes were strongly fragmented in vps8 mutants: the average size of these vesicles strongly decreased relative to the ones in control or rescued animals (*Figure 5A–C,J*, *Figure 5—figure supplement 1*). In addition, the distribution of late endosomes is also altered: Rab7-positive dots almost completely fill the cytoplasm of mutant cells unlike in control or rescued nephrocytes, in which late endosomes form a layer directly beneath peripheral early endosomes (*Figure 5A–C*, *Figure 5—figure supplement 1*). Importantly, the Rbsn-5 positive early endosomal compartment remains unaffected by the absence of Vps8, indicating that early endosomes form but are unable to fuse and generate properly sized late endosomes in mutant cells (*Figure 5A–C,J*, *Figure 5—figure supplement 1*). This model is further supported by the distribution of established lysosomal markers: structures positive for LysoTracker Red, for the lysosomal hydrolase Cathepsin L, and for lysobisphosphatidic acid (LBPA) are all fragmented in vps8 mutants (*Figure 5D–I*). Thus, late endosomes and endolysosomes are fragmented in nephrocytes lacking Vps8. Perhaps as a result of impaired endocytic trafficking, vps8 mutant garland nephrocytes are enlarged compared to wild type or rescued cells (*Figure 5A–C,K*, *Figure 5—figure supplement 1*).

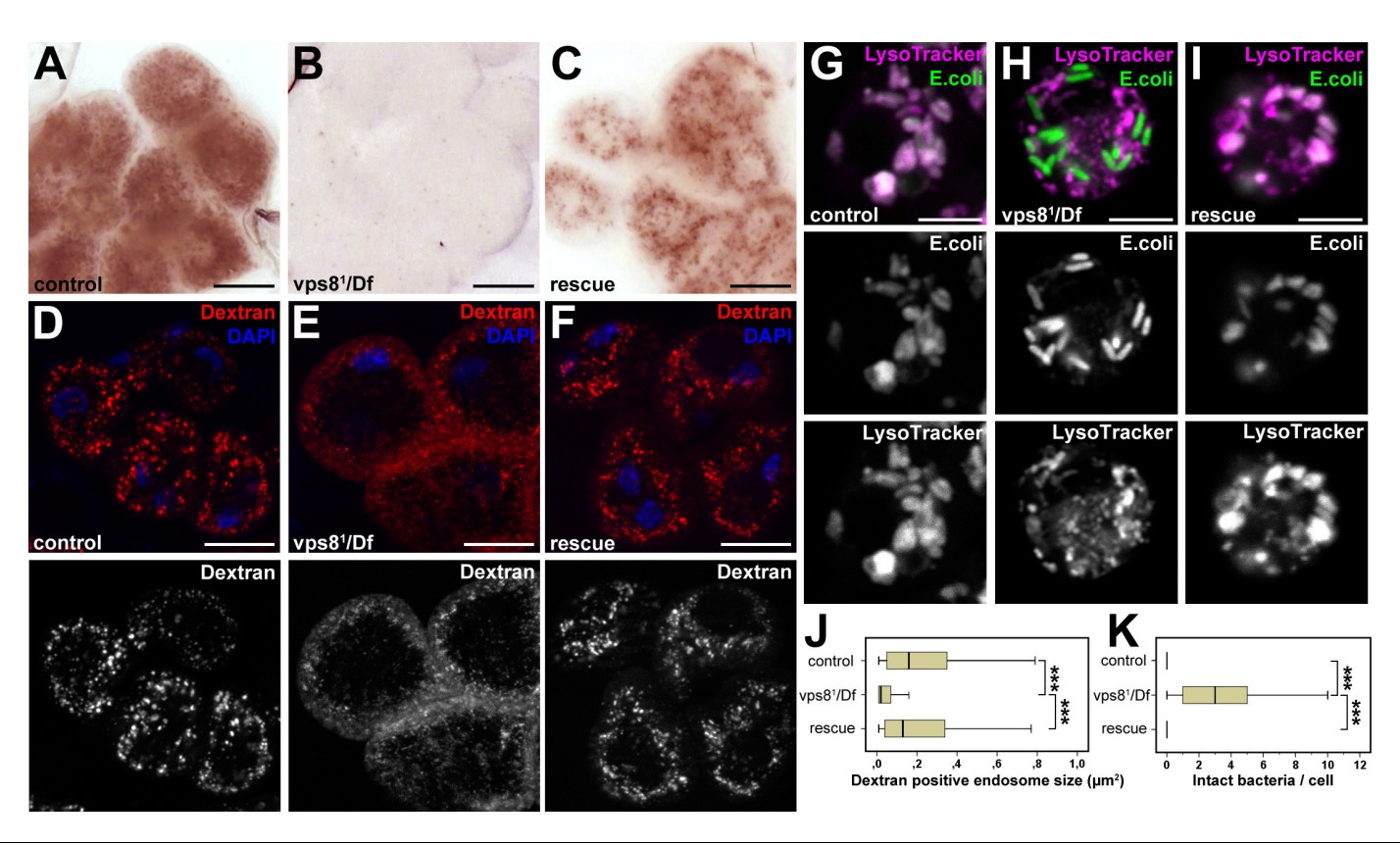

**Figure 4.** Loss of Vps8 impairs endocytic trafficking in garland nephrocytes and hemocytes. (A–C) Garland cells take up silver nitrate from the hemolymph and store it in vesicles, which are visible as brown dots. Compared to control or rescue animals, garland nephrocytes from vps8 mutants contain almost no detectable silver inclusions. (D–F) Vps8 mutant garland cells can take up fluorescent dextran but fail to incorporate it into large endosomes (5 min pulse, no chase). (G–I) Vps8 mutant hemocytes take up FITC labeled bacteria but fail to acidify these endocytic vacuoles. (J,K) Quantification of data from panels (D–F) and (G–I). The medians of data are indicated as vertical black lines within the boxes. Bars show the upper and lower quartiles, ***p<0.001. Bars: (A–F): 20 μm, (G–I): 5 μm.

The following figure supplement is available for figure 4:

**Figure supplement 1.** Loss of Vps8 impairs endocytic trafficking in garland nephrocytes and hemocytes.

Interestingly, the endosomal compartment of vps8 mutant wing imaginal discs is indistinguishable from wild type animals: both the localization of the internalized Notch receptor, and the size and pattern of Rab7-positive late endosomes are similar to controls (*Figure 5—figure supplement 2*). These data raise the possibility that Vps8 acts as a tissue-specific factor with a critical function in endocytic trafficking in a subset of cells, which are highly active in endocytosis (that is, nephrocytes and hemocytes).

## Vps8 forms a complex with Vps16A, Dor/Vps18 and Car/Vps33A

Vps8 is known to form a complex that contains all 4 class C Vps proteins in both yeast and mammalian cells. We immunoprecipitated Vps8-HA from the genomic promoter-driven vps8-HA; vps8[1]/Df adults and larvae, and analyzed the precipitate by western blots. We found that endogenous Car/Vps33A, Dor/Vps18, and Vps16A all interact with Vps8-HA in both adults and larvae (*Figure 6A*, *Figure 6—figure supplement 1*). Moreover, yeast two hybrid experiments showed that Dor/Vps18 directly binds to Vps8 (*Figure 6B*). In line with these, Vps8-HA colocalized with both myc-tagged Dor/Vps18 and endogenous Vps16A in garland nephrocytes (*Figure 6D,E*). Taken together, these data indicate that a CORVET-like complex exists in flies, despite the lack of clear homologs of either

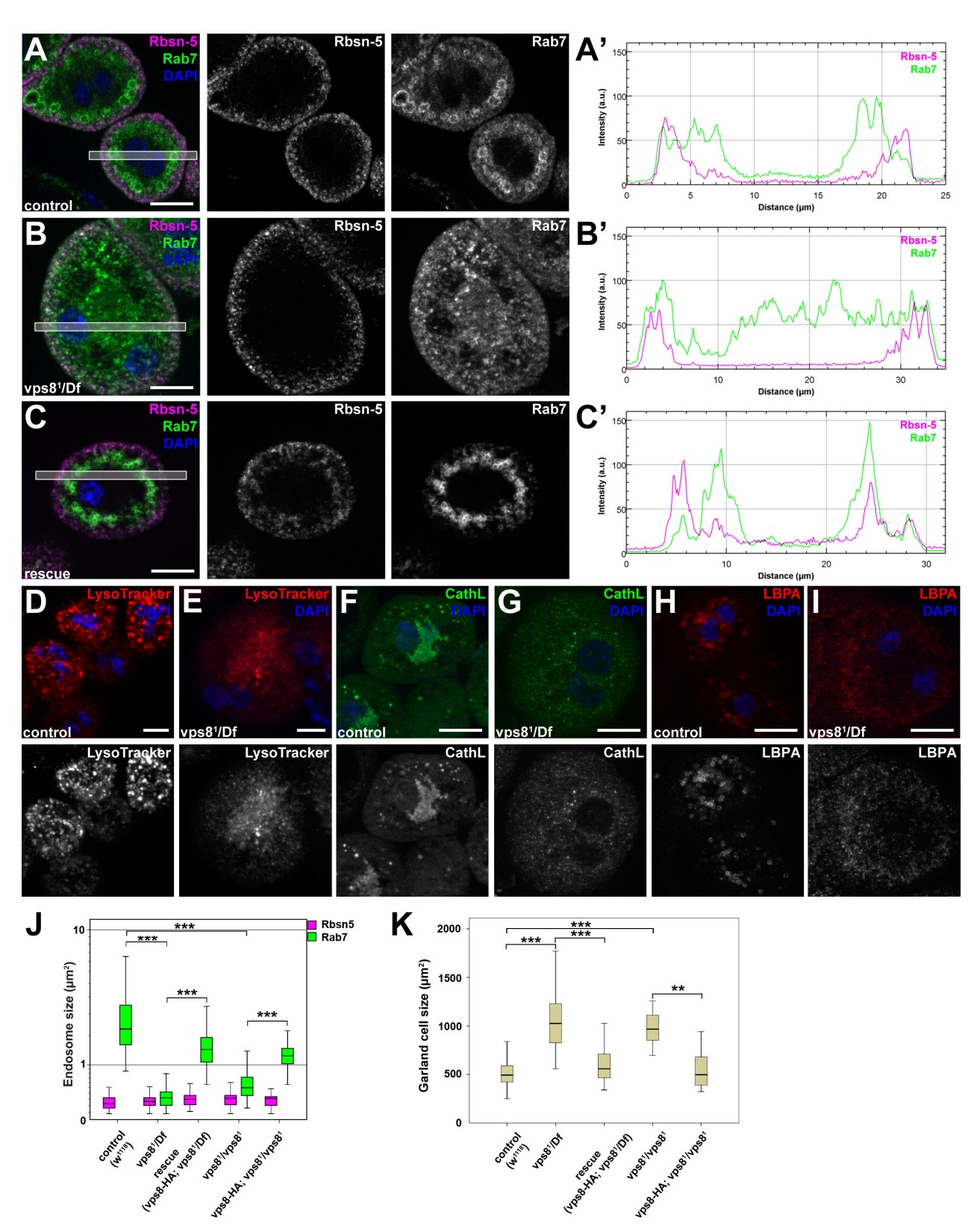

**Figure 5.** The biogenesis of late endosomes and lysosomes is impaired in vps8 mutant garland nephrocytes. (**A–C**) Late endosomes are fragmented in garland nephrocytes lacking Vps8, unlike early endosomes. In wild type control or rescued cells, a layer of Rab7-positive endosomes (green) is found
*Figure 5 continued on next page*

*Figure 5 continued*

under the peripheral Rbsn-5 positive endosomes (magenta). In contrast, small granular Rab7 structures fill the whole cytoplasm of vps8 mutants, while the Rbsn-5 pattern remains unchanged in these cells. Panels (**A'–C'**) show the plot profiles of the framed areas in composite images. (**D,E**) LysoTracker-, (**F,G**) Cathepsin L- (**H,I**) and LBPA-positive lysosomal structures are fragmented in vps8 mutant nephrocytes. (**J**) Quantification of data from panels (**A–C**) and from *Figure 5—figure supplement 1*. (**K**) Vps8 mutant nephrocytes are significantly larger than control or rescued cells. The medians of data are indicated as horizontal black lines within the boxes. Bars show the upper and lower quartiles, **p<0.01, ***p<0.001. Bars: 10 μm. Please see *Figure 5—figure supplements 1,2* for additional data.

The following figure supplements are available for figure 5:

**Figure supplement 1.** The biogenesis of late endosomes is impaired in homozygous vps8 mutant garland nephrocytes.

**Figure supplement 2.** The endosomal compartment of larval wing disc cells is normal in vps8 mutant larvae.

yeast Vps3 or mammalian Tgfbrap1 (*Klinger et al., 2013*; *Li and Blissard, 2015*). In the absence of available antibodies to Drosophila Vps11 and Vps39 proteins, we analyzed the subunit composition of Drosophila CORVET by immunoprecipitation of Vps8-HA as above, followed by tryptic digestion and mass spectrometry. In line with the western blot data, Vps16A, Dor/Vps18, and Car/Vps33A could all be detected as very strong interacting partners of Vps8 (*Figure 6C*), but neither Vps11 nor Vps39 (*Supplementary file 2*). These data suggested that Vps11 might not be a component of this CORVET-like complex in Drosophila that we thus named miniCORVET, consistent with the lack of hemocyte tumors in vps11 mutants (*Figure 3B*).

## MiniCORVET promotes endosomal fusion upstream of HOPS

To extend our studies to other miniCORVET and HOPS subunits, we analyzed early (Rbsn-5-positive) and late (Rab7-positive) endosomes in garland nephrocytes lacking these proteins. As expected, late endosomes are fragmented in cells lacking Dor/Vps18, Vps16A or Car/Vps33A, similar to vps8 mutants (*Figure 7A–C,G*, *Figure 7—figure supplement 1A*), in agreement with the interaction of these proteins. Nephrocytes lacking the HOPS specific subunits Lt or Vps39 showed a very different phenotype: late endosomes were extremely enlarged rather than fragmented, indicating a block of late endosome to lysosome trafficking (*Figure 7E,G*, *Figure 7—figure supplement 1B*). Importantly, we found that both vps11 mutant and RNAi cells have enlarged late endosomes (*Figure 7D*, *Figure 7—figure supplement 1C*) similarly to the loss of HOPS, unlike the fragmentation observed in miniCORVET subunit mutants. This further supports that miniCORVET acts independent of Vps11. Double mutants for vps8 and lt had fragmented Rab7 vesicles (*Figure 7F,G*), indicating that miniCORVET is required for the biogenesis of late endosomes while HOPS promotes their clearance. In all genotypes, early endosomes that carry only Rbsn-5 remained unaffected (*Figure 7A–G*, *Figure 7—figure supplement 1*).

We confirmed these data by electron microscopy. As expected from the Rab7 stainings, large α-vacuoles are present in wild type and rescued animals but absent from vps8 mutant cells, which contain only small endosomes (*Figure 8A–C,I*, *Figure 8—figure supplement 1A,B*), suggesting endosomal fusion defects. Similarly, the endosomal compartment is fragmented and properly sized α-vacuoles are missing from nephrocytes lacking Dor/Vps18, Vps16A, Car/Vps33A, and also from cells double mutant for both vps8 and lt (*Figure 8G–I*, *Figure 8—figure supplement 1C–F*). Of note, these cells contain lots of small electron-dense lysosome-like organelles, which are only seen in nephrocytes lacking both miniCORVET and HOPS function. In contrast with these, the ultrastructure of nephrocytes lacking the HOPS-specific subunits (Lt/Vps41 or Vps39) as well as the class C protein Vps11 is very similar to each other but greatly different from cells lacking miniCORVET. The α-vacuoles of HOPS mutant cells are much larger than those seen in control nephrocytes, and often contain multiple dense cores (*Figure 8D–F,I*, *Figure 8—figure supplement 1G–I*). Multi-core vacuoles with a more electron-dense lumen are also found in the perinuclear region of these cells (*Figure 8—figure supplement 1G–I*). These structures likely arise from the impaired maturation of late endosomes into degradative lysosomes in the absence of HOPS function. Apparently, the formation of such aberrant, enlarged late endosomes and lysosomes requires miniCORVET, because

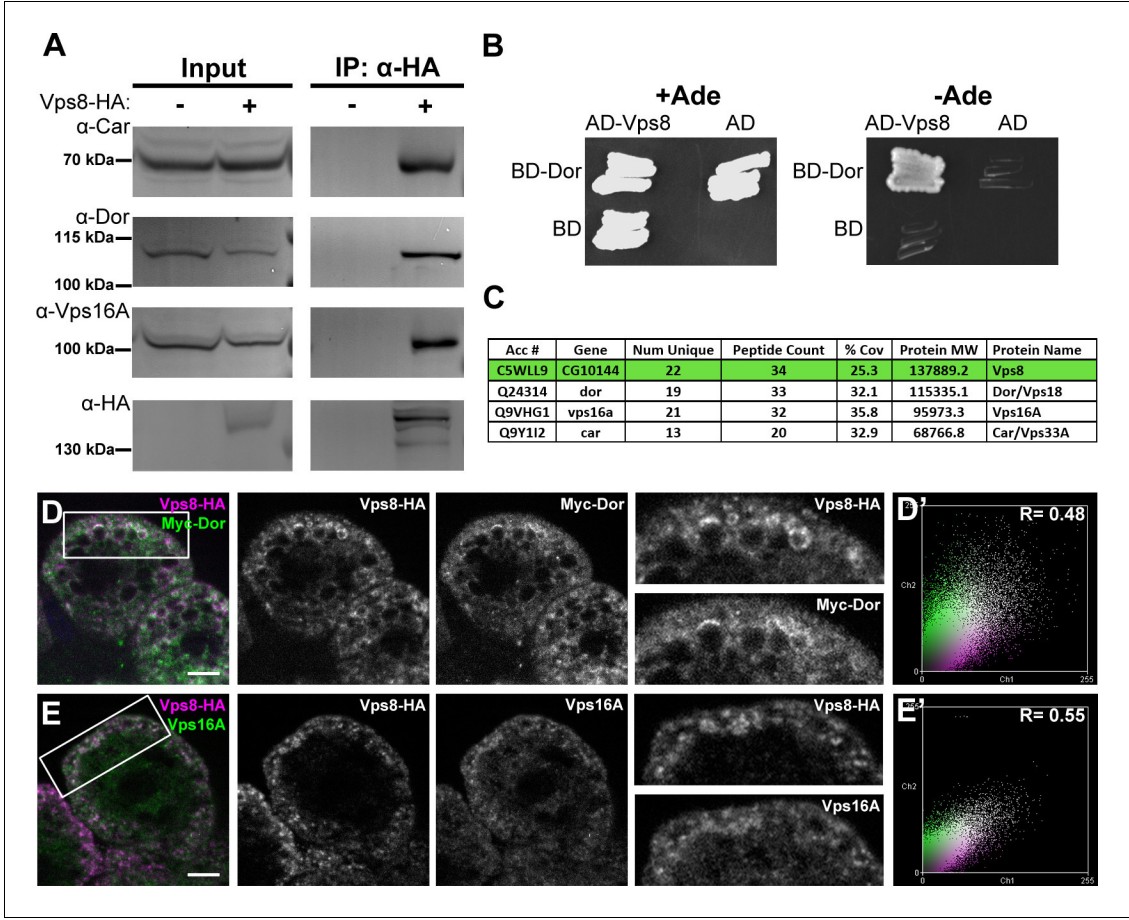

**Figure 6.** Vps8 forms a miniCORVET complex with the class C Vps proteins Dor/Vps18, Car/Vps33A and Vps16A. (**A**) Endogenous Car, Dor and Vps16A proteins coprecipitate with Vps8-HA. (**B**) Vps8 interacts with Dor in yeast two-hybrid experiments. Yeast colony growth on synthetic medium lacking Ade indicates a direct interaction between the two proteins. AD: Gal4 activation domain-containing vector, BD: Gal4 DNA binding domain-containing vector. (**C**) Summary of the mass spectrometry (MS) data from Vps8-HA immunoprecipitates. 3 Vps proteins (Dor, Car, Vps16A) coprecipitated with the bait (Vps8-HA, highlighted with green background) with high peptide count, suggesting that these 4 proteins form a stable complex. Please see **Supplementary file 2** for additional proteomic data. (**D,E**) Vps8-HA colocalizes with myc-tagged Dor (**D**) or endogenous Vps16A (**E**). (**D', E'**) Scatter plots display the intensity correlation profiles of Vps8-HA (magenta) with either myc-Dor or Vps16A (green), with Pearson correlation coefficients shown at the top. Bars: 5 µm.

The following figure supplement is available for figure 6:

**Figure supplement 1.** Vps8 coprecipitates the class C Vps proteins Dor/Vps18, Car/Vps33A and Vps16A from larval lysates.

multiple homotypic endosomal fusion cycles can promote the enlargement of endosomes in HOPS mutants.

## The endosomal localization of Vps8 requires Rab5, Dor/Vps18, Vps16A and Car/Vps33A but is independent of Vps11, Vps39 and Rab7

To understand how miniCORVET is recruited to early endosomes, we analyzed Vps8-HA localization in cells with different genotypes. First we depleted Rab5 or Rab7 in garland nephrocytes. As expected from yeast data, Vps8-HA lost its vesicular localization and became dispersed in the cytoplasm in cells lacking Rab5, but not in cells lacking Rab7 (**Figure 9A,B**, **Figure 9—figure supplement 1A**). We also examined Vps8-HA localization in cells overexpressing the YFP-tagged form of constitutively active Rab5 or Rab7. We found that Vps8-HA localizes to vesicles containing constitutively active Rab5 (**Figure 9C**). On the contrary, Vps8-HA preserved its vesicular localization at the periphery of garland nephrocytes overexpressing active Rab7 (**Figure 9—figure supplement 1B**).

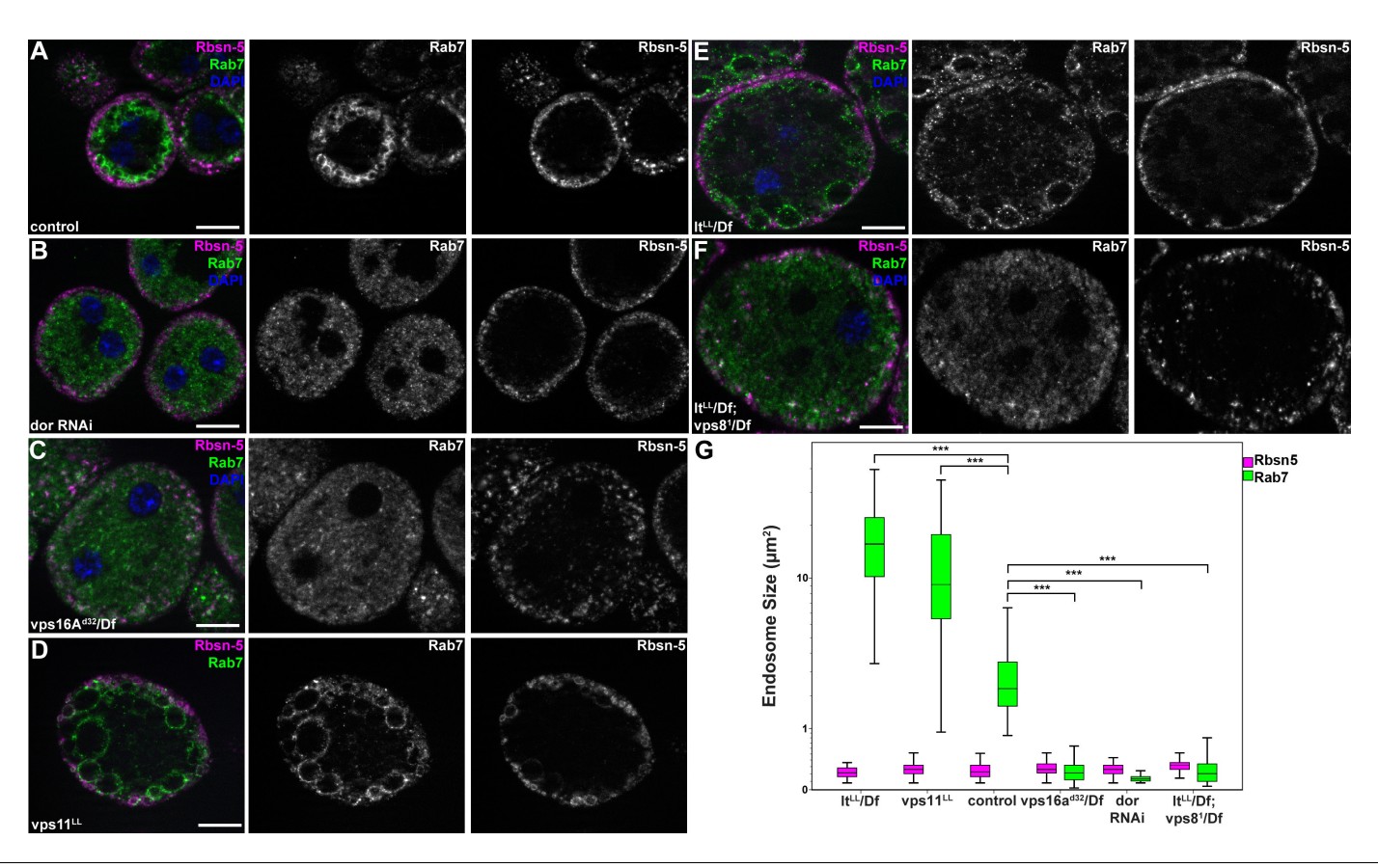

**Figure 7.** MiniCORVET promotes endosomal fusion upstream of HOPS. Garland nephrocytes lacking the miniCORVET subunits Dor/Vps18 (**B**) or Vps16A (**C**) have fragmented Rab7-positive late endosomes compared to controls (**A**), whilst the early endosomal Rbsn-5 signal remains unchanged. In contrast, cells lacking Vps11 (**D**) or the HOPS specific subunit Lt/Vps41 (**E**) have enlarged Rab7-positive vesicles. (**F**) Double mutants for vps8 and lt display a phenotype similar to vps8 mutants, as Rab7-positive late endosomes appear as small dots filling the cytoplasm. (**G**) Quantification of data from panels (**A–F**). The median of data is indicated as a horizontal black line within the boxes. Bars show the upper and lower quartiles, ***p<0.001. Bars: 10 μm. Please see *Figure 7—figure supplement 1*. for additional data.

The following figure supplement is available for figure 7:

**Figure supplement 1.** Additional miniCORVET and HOPS mutant nephrocyte data.

Similar experiments were carried out for Vps8-HA localization in cells lacking miniCORVET or HOPS subunits. We found that the depletion of Dor/Vps18, Car/Vps33A and Vps16A leads to the cytoplasmic dispersion of Vps8-HA (*Figure 9D*, *Figure 9—figure supplement 1C,D*). In contrast, Vps8 sustains its specific early endosomal localization in cells lacking the HOPS subunits Vps39 or Vps11 (*Figure 9E, F*). Importantly, RNAi knockdowns that we used throughout this work resulted in a strong reduction in the level of the corresponding gene products (*Figure 9B*, *Figure 9—figure supplement 2*).

These Vps8-HA localization results provide further evidence that Dor/Vps18, Vps16A and Car/Vps33A function together with Vps8 as part of a miniCORVET complex independent of Vps11 and other HOPS-specific subunits.

## MiniCORVET may cooperate with Rbsn-5 to promote endosomal fusions

As tethers can simultaneously bind two adjacent membranes, it is not necessarily evident how miniCORVET can act as a tether in the absence of the Rab5-binding Vps3/Tgfbrap1 subunit (and also

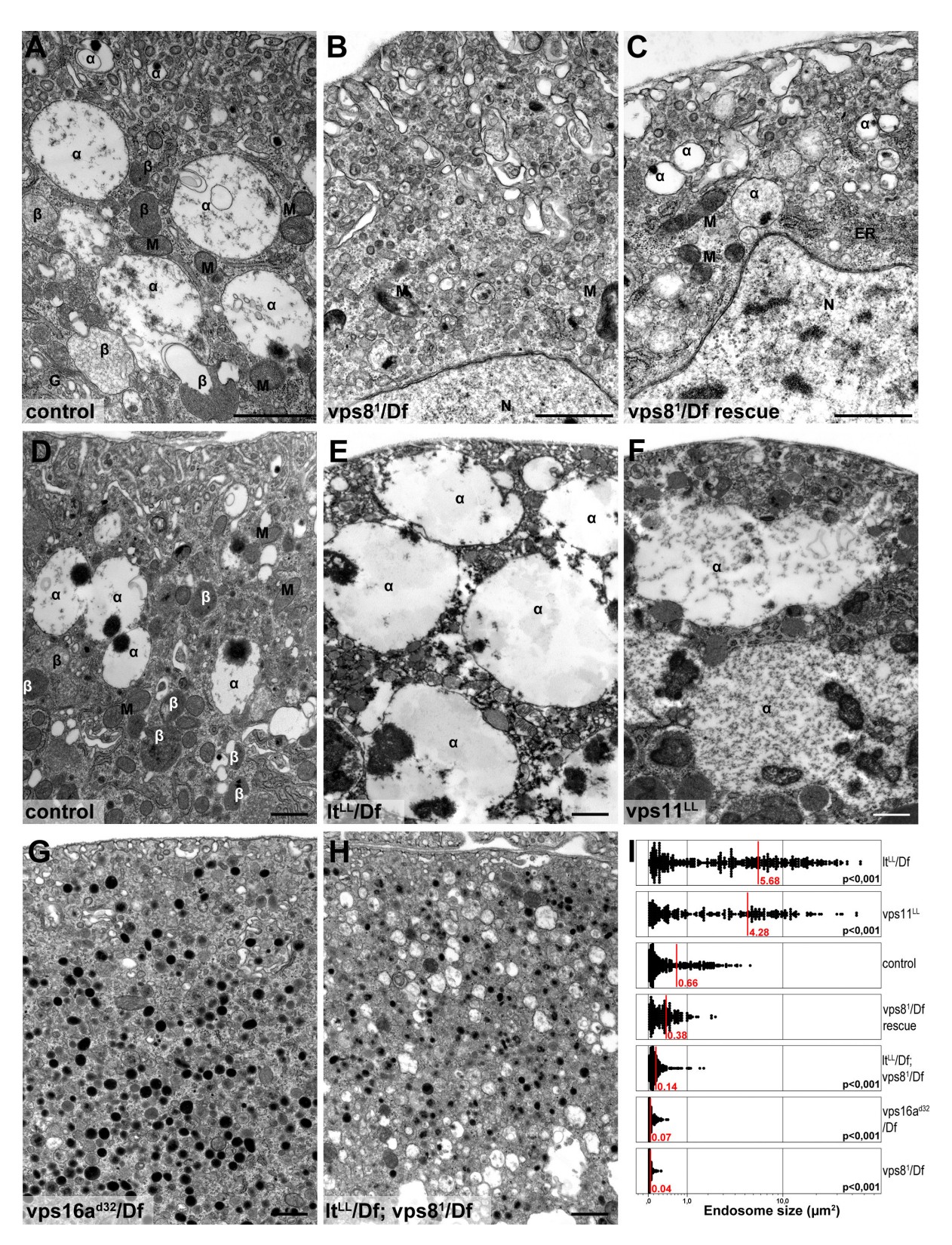

**Figure 8.** Loss of miniCORVET leads to fragmentation of endosomes, unlike HOPS defects. Vps8[1] mutant (**B**) garland nephrocytes lack normal sized α-vacuoles and only small vesicles can be found, unlike in control (**A,D**) or rescued (**C**) cells. In contrast, nephrocytes lacking either the HOPS specific
*Figure 8 continued on next page*

*Figure 8 continued*

subunit Lt/Vps41 (**E**) or Vps11 (**F**) contain enlarged α-vacuoles, which often have multiple cores. The simultaneous loss of both miniCORVET and HOPS function in vps16A single (**G**) or lt and vps8 double mutant cells (**H**) also results in fragmented, small α-vacuoles similar to vps8 single mutants. Note that these nephrocytes also contain numerous small electron-dense organelles, which may represent a Golgi-derived pre-lysosomal compartment. N: nucleus, M: mitochondria, ER: endoplasmic reticulum. (**I**) Quantification of data from panels (**A–G**). The size of each endosome is represented as a single punctum on the dot plots. Red lines and numbers show the mean endosome area in μm². Bars: 1 μm. Please see *Figure 8—figure supplement 1.* for additional data.

The following figure supplement is available for figure 8:

**Figure supplement 1.** Additional miniCORVET and HOPS mutant nephrocyte data.

Vps11) from one end of the complex. Considering the fragmented endosomal phenotype, we hypothesized that miniCORVET may cooperate with a partner on early endosomes, such as the other early endosomal tether Rbsn-5. First, miniCORVET colocalizes with Rbsn-5 (*Figure 2E*). Second, yeast data showed that class C Vps proteins, including Vps18p/Pep3p, interact with Vac1p/Pep7p, the yeast homolog of Rbsn-5 (*Peterson and Emr, 2001*; *Srivastava et al., 2000*). We therefore analyzed the potential binding of Rbsn-5 to either Dor/Vps18 or Vps8 in yeast two hybrid experiments, and detected its strong direct interaction with Dor/Vps18 (*Figure 10A*). Importantly, nephrocytes lacking Rbsn-5 also contained fragmented Rab7-positive endosomes and were devoid of properly sized α-vacuoles based on ultrastructural analysis (*Figure 10B,C*). These phenotypes are remarkably similar to that of vps8 mutants. Finally, we analyzed the localization of Vps8-HA in cells lacking Rbsn-5. Rab5-positive early endosomes were fragmented and accumulated in the perinuclear region in the absence of Rbsn-5, but these endosomes still contained Vps8-HA (*Figurre 10D*). These results indicate that Rbsn-5 also acts as an essential early endosomal tether in garland nephrocytes. As cells lacking Vps8 have a normal Rbsn-5 positive endosomal compartment (*Figure 5A,B*), the endosomal recruitment of Rbsn-5 and miniCORVET appears to be mutually independent of each other. Strikingly, the formation of properly sized late endosomes and endolysosomes in nephrocytes thus requires the function of both Rbsn-5 and miniCORVET, which could be explained by their cooperative interaction.

## Discussion

CORVET is an early endosomal tether in yeast. The 4 class C Vps proteins (Vps11, 16, 18, 33) form its core that is shared with the HOPS complex, and the 2 specific subunits Vps3 and Vps8 are located on the opposite sides of the complex. Instead of these 2 proteins, the late endosomal tether HOPS contains Vps39 and Vps41. The specific subunits are thought to be important for the bridging of appropriate vesicles, since both complexes act as a tether during vesicle fusions: CORVET is associated with Vps21/Rab5-positive early endosomes, while HOPS localizes to Ypt7/Rab7 containing late endosomes and the vacuole/lysosome, respectively (*Balderhaar and Ungermann, 2013*; *Bröcker et al., 2012*; *Plemel et al., 2011*; *Solinger and Spang, 2013*). Most of these proteins are conserved in higher eukaryotes, and the composition and function of HOPS is similar to the yeast complex in metazoan cells. CORVET is less well characterized: it contains the specific subunits Vps8 and Tgfbrap1, and associates with a subset of endosomes in HeLa cells (*Lachmann et al., 2014*; *Perini et al., 2014*).

Our data clearly show that a CORVET-like complex is a functional and essential early endosomal tether in a subset of cell types in Drosophila, that is, nephrocytes and hemocytes. Importantly, it is these cells that probably display the highest endocytic activity in the larva. The endolysosomal system occupies much of the cytoplasm in garland nephrocytes, and based on ultrastructural studies, their unusually large late endosomes and lysosomes even deserved a name of their own: α-vacuoles and β-vacuoles, respectively (*Koenig and Ikeda, 1990*; *Kosaka and Ikeda, 1983*). Flies with mutations in genes whose products are required for early endosomal trafficking, such as avl/Syntaxin 7 (*Lu and Bilder, 2005*), Rab5 (*Wucherpfennig et al., 2003*) and Rbsn-5 (*Morrison et al., 2008*), die during embryonic development. In contrast, vps8 mutant flies are semilethal. Importantly, the structure and pigmentation of their compound eye and the endosomal compartment of mutant wing disc

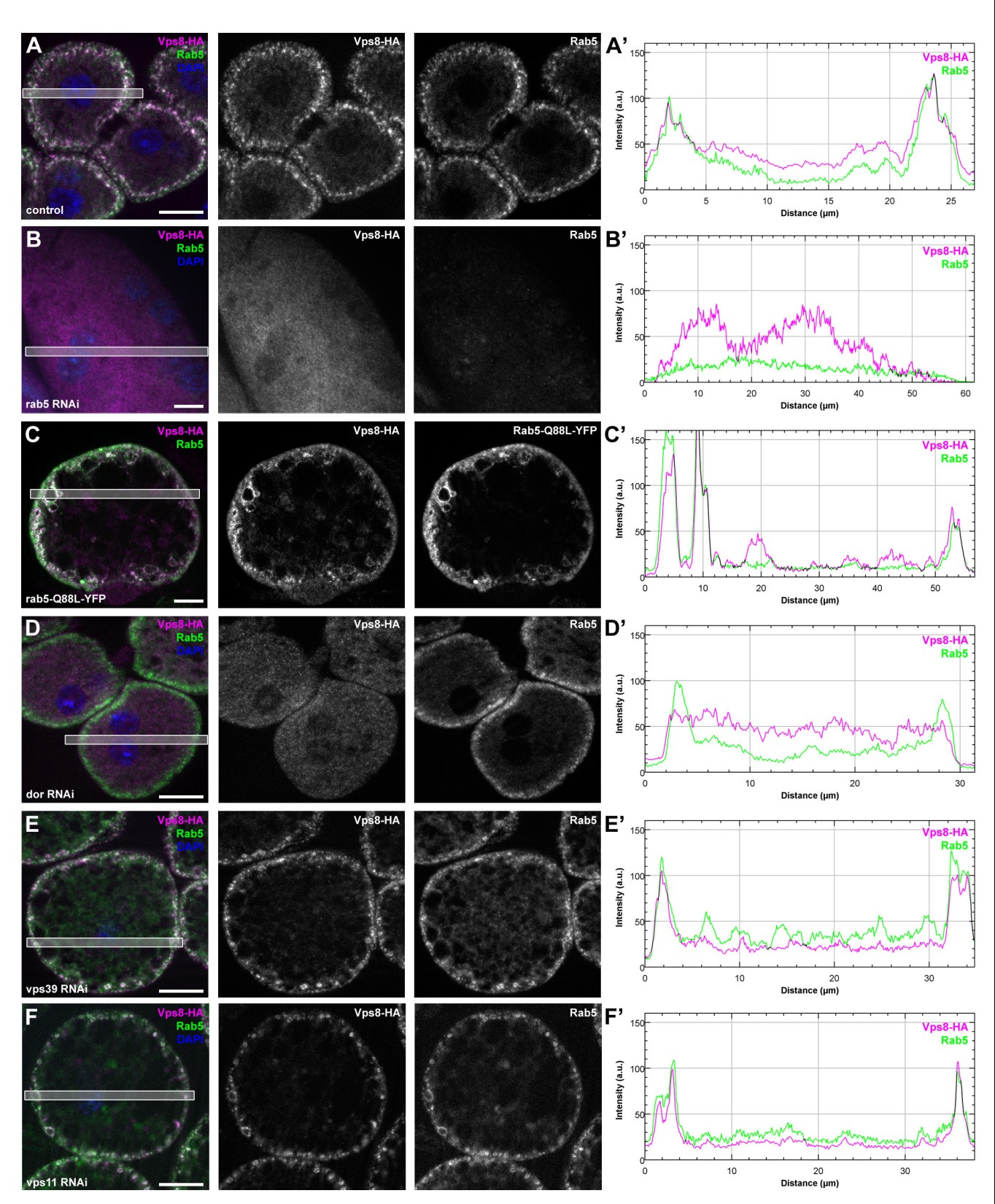

**Figure 9.** Vps8 localization to early endosomes depends on Rab5 and miniCORVET subunits, but not the HOPS complex. **(A–F)** Images from garland nephrocytes expressing Vps8-HA in different genetic backgrounds (as indicated) were stained with anti-HA (magenta) and anti-Rab5 (green). Plot

*Figure 9 continued on next page*

*Figure 9 continued*

profiles of the framed areas are shown in panels (**A′–F′**). Vps8-HA and Rab5 colocalize at the periphery of control cells (**A**). In the absence of Rab5, Vps8-HA is dispersed in the cytoplasm. Note the absence of Rab5 signal, which indicates efficient RNAi knockdown. (**B**). The colocalization of Vps8-HA with Rab5 is maintained on enlarged early endosomes in cells expressing the constitutively active form of Rab5 (**C**). Rab5-positive early endosomes are present at the periphery of nephrocytes in the absence of Dor, but Vps8-HA is no longer found on these structures (**D**). In contrast with this, loss of either Vps39 (**E**) or Vps11 (**F**) does not affect the localization of Vps8-HA to Rab5-positive early endosomes. Bars: 10 μm. Please see *Figure 9—figure supplement 1*. for additional data.

The following figure supplements are available for figure 9:

**Figure supplement 1.** The early endosomal localization of Vps8 is independent of Rab7 and requires Car/Vps33A and Vps16A.

**Figure supplement 2.** Validation of knockdown efficiencies for RNAi lines used in this study.

cells remains similar to wild type ones. Late endosomes and likely as a consequence endolysosomes are fragmented in vps8 mutant nephrocytes, and bacteria-containing phagosomes fail to mature into acidic phagolysosomes in hemocytes. These results raise the possibility that CORVET activity is most critical in nephrocytes and hemocytes in Drosophila. It remains to be established whether such remarkably strong phenotypes are also seen in the mammalian kidney and white blood cells in the absence of Vps8.

Rbsn-5 is another early endosomal tether in Drosophila cells (*Morrison et al., 2008*), and we show that it is also required for the generation of properly sized late endosomes and endolysosomes in garland nephrocytes. Based on our vps8 mutant data, the early endosomal tethering activity of Rbsn-5 might be sufficient for endocytic flux in most cells, with the exception of nephrocytes and hemocytes. This is likely due to the very high endocytic activity of these cells, and perhaps also to their unusually large endosomes and phagosomes, as the decreased membrane curvature may require the contribution of a different tether as well. We speculate that in this case, Rbsn-5 may cooperate with a 4-subunit miniCORVET complex through direct binding to Dor, an evolutionarily conserved interaction that was reported earlier between the corresponding yeast homologs, Vac1p/Pep7p and Vps18p/Pep3p (*Srivastava et al., 2000*). This is further supported by the colocalization of Rbsn-5 with Vps8. Our model would also explain how miniCORVET could act as a tether in Drosophila in the absence of Vps3 and Tgfbrap1 homologs. Importantly, the recruitment of Vps8 and Rbsn-5 to Rab5-positive early endosomes is mutually independent of each other, suggesting that they interact after associating to the surface of these vesicles.

Perhaps the most surprising aspect of our study is that Vps8 appears to act independent of Vps11 in Drosophila. Several lines of evidence support this finding. First of all, the endosomal compartment is fragmented in vps8 mutants as well as in the absence of Vps16A, Dor/Vps18 and Car/Vps33A, whereas loss of the HOPS-specific subunits Vps39 and Lt/Vps41 lead to the enlargement of late endosomes and endolysosomes, a phenotype that is also seen in Vps11 mutant and RNAi cells. A summary of the phenotypes of cells lacking various miniCORVET and HOPS subunits is shown in *Figure 11*. Second, Vps11 is dispensable for the recruitment of Vps8 to Rab5-positive early endosomes similar to Vps39, while the loss of Rab5, Dor, Car or Vps16A all result in cytoplasmic dispersal of Vps8. Third, our proteomic experiment identified Car, Dor, and Vps16A as interacting partners of Vps8 with 13–21 unique peptides for each one, suggesting that these proteins form a stable complex. In contrast, Vps11 was not identified in this immunoprecipitate. Our data go along with those of a recent pulldown study, which identified Vps8, Car, Dor and Vps16A in a search for Rab5 effectors in Drosophila cells, but not Vps11, Vps39 or Lt (*Gillingham et al., 2014*). Importantly, Vps11 was found in a Rab2 pulldown experiment along with the other 5 HOPS subunits, unlike Vps8 (*Gillingham et al., 2014*). These independent data further support the existence of a full HOPS and the novel, 4-subunit miniCORVET complex in Drosophila.

HOPS and CORVET can be divided into two separate parts: a head part composed of the class C proteins Vps33, Vps16, Vps18, and the CORVET or HOPS specific proteins Vps8 or Vps41, respectively. The tail part contains the class C protein Vps11, and Vps3 or Vps39 for CORVET or HOPS, respectively (*Bröcker et al., 2012*; *Ostrowicz et al., 2010*; *van der Kant et al., 2015*). An approximately 200 residue region near the Vps11 C-terminus is necessary and sufficient to hold Vps3 or

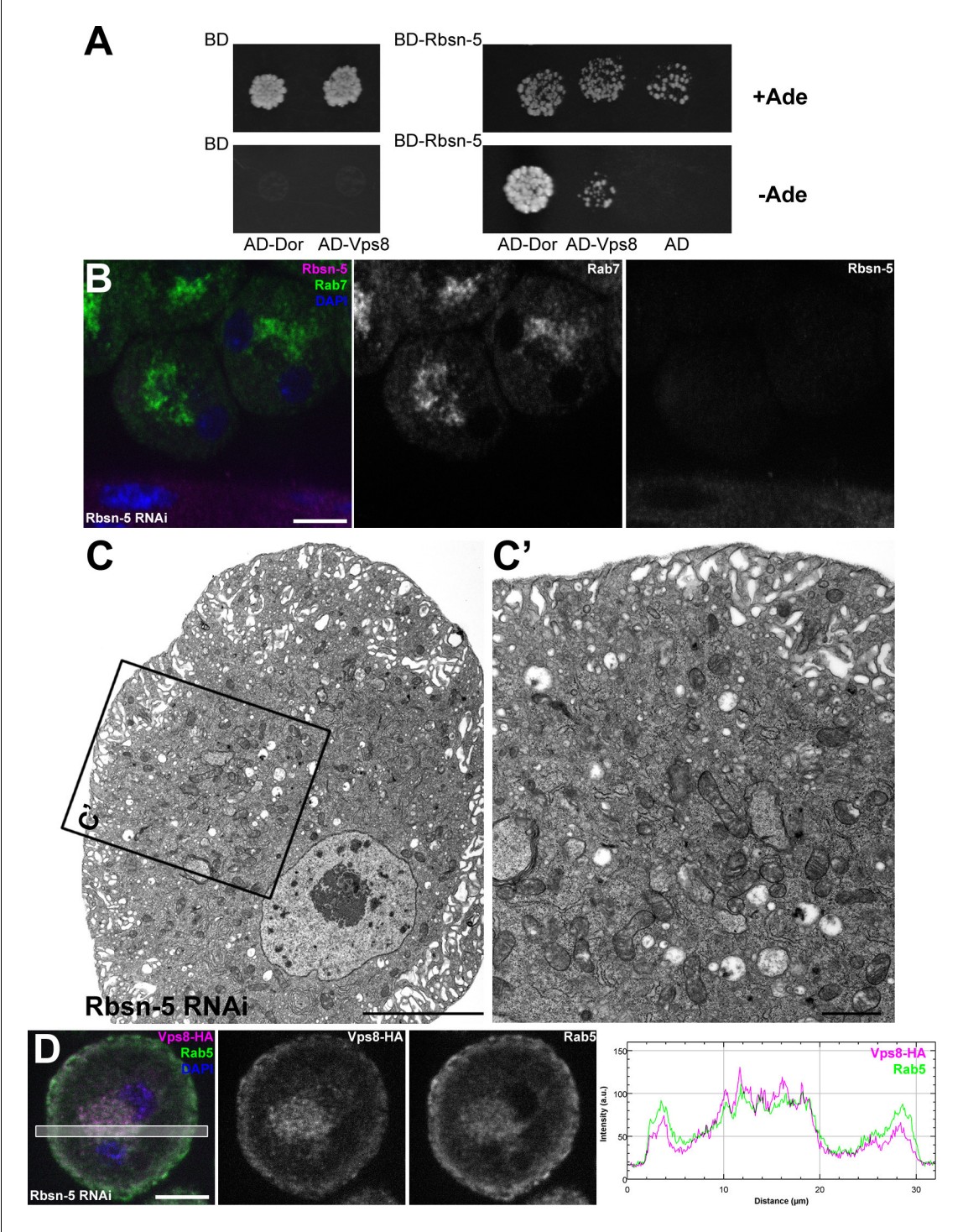

**Figure 10.** Rbsn-5 binds to Dor/Vps18 and is also required for early endosomal fusions. (A) Rbsn-5 directly binds to Dor, based on prominent growth of yeast colonies on synthetic medium lacking Ade in yeast two hybrid experiments. (B) Rab7-positive late endosomes are fragmented in cells undergoing Rbsn-5 RNAi. Note the absence of Rbsn-5 signal in nephrocytes but not in the gut that is visible in the bottom part of this panel, indicating efficient knockdown. (C) Endosomes are fragmented in Rbsn-5 RNAi garland cells, based on the lack of large α-vacuoles in ultrastructural images. (D) In the absence of Rbsn-5, Rab5-positive early endosomes are fragmented and accumulate in the perinuclear region of garland nephrocytes, and the plot profile of the framed area is also shown. Note that Vps8-HA remains associated with these early endosomes. Bars: (**B**,**D**): 10 μm, (**C**,**C'**): 5 and 1 μm, respectively.

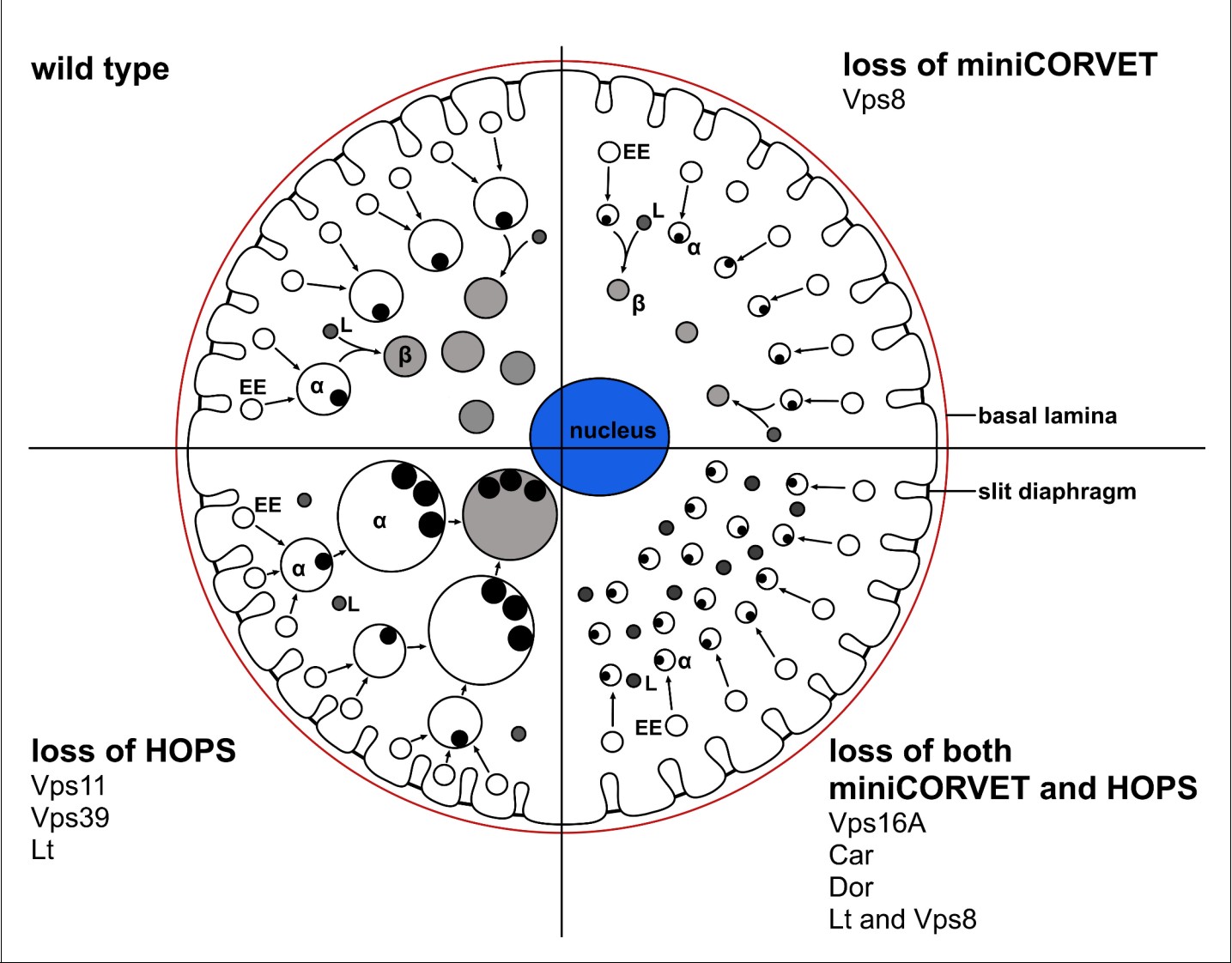

**Figure 11.** A cartoon illustrating garland nephrocytes in wild type, miniCORVET and/or HOPS loss-of-function animals. In control cells, fusion of early endosomes leads to formation of large electron lucent late endosomes (α-vacuoles) that contain a single core. These vacuoles mature into degradative endolysosomes (β-vacuoles), possibly by fusing with primary lysosomes. In cells lacking Vps8 (and as a result miniCORVET function), α-vacuoles remain small and fragmented due to insufficient early endosomal fusion events. In cells lacking HOPS function due to loss of Lt, Vps39 or Vps11, late endosomes fail to mature into degradative lysosomes. Note that α-vacuoles enlarge into enormous vacuoles containing multiple dense cores in these cells, likely because of the continuous input from early endosomal fusions that are supported by the tethers miniCORVET and Rbsn-5. Nephrocytes lacking both miniCORVET and HOPS function (due to loss of the shared subunits Vps16A, Car, Dor, or the double mutation of Vps8 and Lt) contain fragmented late endosomes as in the case of Vps8 mutants, and accumulate small dense granules (possibly Golgi-derived vesicles transporting lysosomal cargo). EE: early endosome, L: primary lysosome.

Vps39 onto the rest of the class C Vps core in yeast (*Ostrowicz et al., 2010*; *Plemel et al., 2011*). This observation is compatible with our model of Vps11 being dispensable for miniCORVET assembly in the absence of Vps3. We thus propose that unlike a fully formed HOPS complex, a miniCORVET corresponding to the head part of the yeast or mammalian CORVET complexes exist in Drosophila. In addition to its tethering function, recruitment of the SM (Sec1/Munc18) protein Car/Vps33A may also facilitate the fusion of early endosomes by acting as a template for SNARE assembly (*Baker et al., 2015*).

Taken together, we identify the novel 4-subunit miniCORVET complex as a tether for early endosomes that is essential for endocytic trafficking in nephrocytes and hemocytes in Drosophila, and show that it acts upstream of HOPS, which mediates the fusion of lysosomes with late endosomes.

## Materials and methods

### Fly stocks, culture and silver uptake experiments

Flies were raised at 25°C in standard conditions. Prospero-Gal4 (pros-Gal4, gift of Bruce Edgar, ZMBH Heidelberg, Germany) was used to drive upstream activation sequence (UAS) dependent transgene expression in garland nephrocytes (*Bechtel et al., 2013*). The following Gal4 responsive transgenic lines were used: Rab5 RNAi [JF03335], Rbsn-5 RNAi [HMC04769], UAS-YFP-Rab5-Q88L, UAS-YFP-Rab7-Q67L, Act-Cas9 - obtained from Bloomington Drosophila Stock Center (BDSC, Bloomington, IN), and Rab7 RNAi [GD40337], Vps16A [GD13782], car [GD1397], dor [KK102176], Vps39 [GD12152], Vps11 [KK102566] - obtained from Vienna Drosophila Resource Center (VDRC, Vienna, Austria). For colocalization experiments in garland cells expressing Rab7-GFP or Lamp1-GFP, larvae carrying Vps8-HA/hs-Flp; UAS-Dcr2; Actin>CD2>Gal4 UAS-Rab7-GFP or UAS-LAMP1-GFP were used (*Takáts et al., 2014*). Mutant lines used in this study include lt[LL07138] (*Schuldiner et al., 2008*), vps11[LL06553], vps16a[d32] (*Takáts et al., 2014*), w[1118], and the deficiencies Df(2L)lt45, Df(3R)ED5339, Df(3L)ED211 (obtained from BDSC). For ubiquitous expression of RNAi constructs we used either Act5C-Gal4 or tub-Gal4 obtained from BDSC. Hml-Gal4, UAS-GFP was used to visualize the sessile hematopoetic compartment (*Csordás et al., 2014*). The experimental genotypes that we analyzed are shown in *Supplementary file 3*. Larvae and adult animals were photographed on a Lumar V12 stereomicroscope equipped with AxioCam ICc camera (Carl Zeiss, Jena, Germany).

For silver uptake experiments adult flies were allowed to lay eggs onto 0.005% AgNO$_3$ containing minimal medium (20 g yeast in 35 ml 1.5% agar). Silver free medium was used for control experiments. Progeny were allowed to develop until the wandering L3 larval stage, when garland nephrocytes were dissected in PBS and fixed with 4% formaldehyde in PBS (30 min at room temperature), and then photographed. Images from multiple focal planes were captured with AxioCam ICc camera on an AxioImager Z1 microscope using a Plan-Neofluar 40x/0.75 NA objective (Carl Zeiss). Images were then projected onto one composite image with CombineZP software. Before taking the photographs, all garland nephrocytes were UV illuminated for 10 s using the microscope's HBO 100 lamp and DAPI filter set (#49), as we initially observed that a short ultraviolet illumination permanently turns the yellowish color of the intracellular silver inclusions into brown.

### Quantitative real-time PCR (qPCR)

RNA samples were prepared from pools of 10–20 mg wandering larvae per genotype using Direct-zol RNA MiniPrep Kit (Zymo Research, Irvine, CA), which was followed by cDNA synthesis using High-Capacity cDNA Reverse Transcription Kit (Applied Biosystems) according to the manufacturer's instructions. QPCR reactions were performed on a Rotor-Gene Q instrument (QIAGEN, Valencia, CA) with gene-specific primers using Rotor-Gene SYBR Green Kit (QIAGEN) following the manufacturer's instructions. The following primers were used: AAAAGCTTACAAAATGTGTGACGA and CAATCGATGGGAAGACGG for Actin5C; GATCTTCGAGCACCTTCCC and CATCTGCATCTCGTGCTTGT for Vps39; GAAGGGGTCCTTGTTTTGCT and AGTGTCGTTGCATTCTGCTTT for Vps11. The following primer pairs were used for Vps8: TAAGCTTCCTGGTAAAACACCATC and GATACGCTCCTTGGCTTGAG (frameshift independent primers), and TCTTGATGATGCCGAGTTTG and GAATCCGCCTCAAATTCACT (frameshift sensitive primers). All SYBR green assays were performed in triplicates. Relative expression ratios were calculated as ratios normalized to Actin5C.

### Construction of Vps8-HA transgenic and vps8 mutant D. melanogaster lines

To construct HA tagged Vps8 expressed by endogenous vps8 promoter sequences, we first generated a pGen-9xHA vector by replacing the Acc65I-EcoRI mCherry coding fragment of the previously described pGen-Cherry vector (*Takáts et al., 2014*) with a 9×HA coding sequence using annealed synthetic oligos. Next, the genomic region containing Drosophila CG10144 was amplified using primers TATAAATCGCCTGGGCGGTGC and CACGCACACCCGCACACATAC (plus 5' overhangs

containing desired restriction sites) and cloned into pGen-9xHA as a NotI-Acc65I fragment. Transgenic Vps8-HA flies were generated by random insertion using w[1118] embryos and standard procedures (BestGene).

To generate a pair of gRNAs that target the CG10144/vps8 locus, we carried out a PCR (primers: TATATAGGAAAGATATCCGGGTGAACTTCGGTGGGCGGTATTGCAAACTGTTTTAGAGCTAGAAA TAGCAAG and ATTTTAACTTGCTATTTCTAGCTCTAAAACGCATTGACACGTTCCGCCGCGACG TTAAATTGAAAATAGGTC) using pCFD4 vector as template (Port et al., 2014). Next, the resulting fragment was cloned into the pCFD4 vector by using Gibson Assembly kit (New England BioLabs, Ipswich, MA). This construct was injected into y[1] v[1] nos-phiC31\int.NLS; attP40 embryos to integrate the DNA onto the second chromosome, carried out by the Drosophila injection core facility at the Biological Research Centre in Szeged, Hungary. The transgenic flies carrying this double gRNA construct were crossed to Act-Cas9 flies, followed by the construction of independent descendant lines that were screened for mutations by PCR and DNA sequencing using primers: CGAAAATGGAA TGGCCTAGA and CAGCTTCCGGGTGAAACTAA.

## Yeast two-hybrid assay

Vps8, Rabenosin-5, were amplified using the following primer pairs (and templates): vps8: ATG TCGGAGCTTAAGGCCCCGT, CTATATAAATCGCCT GGGCGGTGC (EST: AT14809); rbsn5: ATGTC TGGAAATCCTTTCGACAGC, TAGGCCTCTGTGGGTGAGGCG (EST: LD29542); dor: ATGGACACG TCTATGCCTAACCAGC, CTACTCCCACTCGACATTCACCTGC (UAS-myc-Dor) (Sevrioukov et al., 1999) containing appropriate 5' overhangs with desired restriction sites. The fragments were cloned into pGADT7 AD (Gal4-DNA activation domain) and pGBKT7 BD (Gal4 DNA binding domain) vectors (Clontech, Mountain View, CA) and then transformed into the yeast strain PJ69-4A using the Frozen-EZ Yeast Transformation II kit (Zymo Research). The transformants were selected by growth in minimal medium (Trp−, Leu−), and to assay for activation of the reporter gene and hence interaction, transformants were selected by growth on Trp−, Leu−, Ade− plates. Empty vectors were used for negative controls. At least three colonies were checked for interaction for each transformation.

## Western blot, co-immunoprecipitation and mass spectrometry

Hemocytes were collected from L3 larvae by bleeding into Schneider's insect medium (Sigma Aldrich, Budapest, Hungary) supplemented with 0.01% 1-phenyl-2-thiourea (Sigma Aldrich, P7629) and 5% FBS on ice, and centrifuged at 1700 rpm for 8 min at 4°C. Pelleted cells were resuspended in 1x Laemmli solution. Protein samples of hemocytes and whole L3 larvae were separated by SDS-PAGE on a 10% polyacrylamide gel and transferred to PVDF membrane as described (Takáts et al., 2013). After blocking with 0.5% casein, monoclonal rat anti-HA (1:2000; Roche, Basel, Switzerland), monoclonal mouse anti-tubulin (1:2000; AA4.3-s; Developmental Studies Hybridoma Bank, DSHB, Iowa City, Iowa) primary and alkaline-phosphatase-conjugated anti-rat and anti-mouse (1:5000; EMD Millipore - Merck Hungary, Budapest) secondary antibodies were used as before (Takáts et al., 2013).

For co-immunoprecipitation experiments, 220 mg of adults or larvae (w[1118] as control, vps8-HA;+;vps8[1]/Df as vps8-HA) were frozen at -80°C, followed by homogenization in liquid nitrogen using a pre-chilled mortar and lysed in lysis buffer (0.25% Triton X-100, 150 mM NaCl, 1 mM EDTA and 20 mM Tris-HCl, pH 7.5) containing protease and phosphatase inhibitors (1 mM PMSF, 10 µg/ml leupeptin, 10 µg/ml aprotinin, 0.5 mM NaF, 0.5 mM $Na_3VO_4$). Lysates were spun twice at 20,000 g for 10 min at 4°C in a 5430R centrifuge (Eppendorf, Hamburg, Germany) to completely get rid of fat and unbroken cuticle pieces. The cleared supernatant was supplemented with 20 µl anti-HA slurry (Sigma Aldrich). After incubation on a rotator at 4°C for 4 hr, beads were collected by centrifugation at 5000 g for 30 s at 4°C, followed by extensive washes in lysis buffer and finally boiled in 25 µl Laemmli sample buffer. For western blot the following antibodies were used: rabbit anti-Car 1:1000 (Akbar et al., 2009), rabbit anti-Dor 1:1000, rabbit anti-Vps16A 1:2000 (Pulipparacharuvil et al., 2005), monoclonal rat anti-HA 1:2000 (Roche). Anti-Car, anti-Vps16A, anti-Dor antibodies were gifts of Helmut Krämer (UT Southwestern Medical Center, USA). Secondary antibodies were: alkaline phosphatase-conjugated anti-rabbit and anti-rat (both 1:5000; Millipore). Blots were developed by exposing the membrane to an NBT/BCIP colorimetric substrate solution (VWR, Debrecen, Hungary).

For MS experiments 1.5 g vps8-HA;vps8[1]/Df and w[1118] (as control) adult flies were treated as in the co-immunoprecipitation experiments with the following modifications: the flies were homogenized in lysis buffer containing 50 mM Tris-HCl (pH = 7.5), 150 mM NaCl, 0.5 M EDTA, 0.5% TritonX-100. 1 mM PMSF, 10 μg/ml leupeptin, 10 μg/ml aprotinin, 0.5 mM NaF, 0.5 mM $Na_3VO_4$ and cleared as above. The lysates were incubated with 100 μl monoclonal anti-HA agarose beads (Sigma Aldrich) for 2 hr at 4°C and then collected by centrifugation at 5000 g for 2 min at 4°C. After intensive washes with wash buffer (lysis buffer without detergent and inhibitors), precipitated proteins were eluted with 6M Guanidin-HCl. Next, proteins were digested with trypsin as follows: protein disulfide bridges were reduced using dithiothreitol (2 μl 100 mM DTT / 25 mM ammonium bicarbonate (ph = 7.5) (ABC) and the free sulfhydryl groups were alkylated using iodoacetamide (2.2 μl 200 mM DTT / 25 mM ABC). The resulting reaction mixture was concentrated on 30 kDa MWCO filter units, washed 3 times with 200 μl 6 M guanidine hydrochloride / 25 mM ABC then 3 times with 200 μl 25 mM ABC and treated with trypsin (1 μg sequencing grade modified trypsin (Promega, Madison, WI) / 100 μl 25 mM ABC). The resulting peptide mixtures were isolated by centrifugation and dried down, and redissolved in 20 μl 0.1% formic acid (FA) / water. ¼ of the samples were subjected to LC-MS/MS analyses using a nanoAcquity UPLC (Waters, Budapest, Hungary) on-line coupled to an Orbitrap-Elite mass spectrometer (Thermo Fisher, Waltham, MA) operating in the positive ion mode. After trapping at 3% B (Waters Symmetry C18 180 μm x 20 mm column, 5 μm particle size, 100 Å pore size; flow rate: 10 μl/min), peptides were separated using a linear gradient of 10–40% B in 90 min (Waters BEH300 C18 75 μm x 250 mm column, 1.7 μm particle size, 300 Å pore size; flow rate: 300 nl/min); solvent A was 0.1% FA/water, and solvent B was 0.1% FA/5% dimethyl sulfoxide acetonitrile. Data acquisition was carried out in a data-dependent fashion, the 10 most abundant, multiply charged ions were selected from each MS survey scan (m/z: 380–1600). Precursor masses were measured in the Orbitrap, and CID spectra were acquired in the linear ion trap (normalized collision energy: 35). Raw data were converted into peak-lists using the PAVA software (*Guan et al., 2011*) and searched with the Protein Prospector search engine (v.5.14.1.) applying the following parameters: mass accuracy: 5 ppm for precursor ions and 0.6 Da for fragment ions (both specified as monoisotopic values); enzyme: trypsin with maximum 1 missed cleavage site; fixed modification: carbamidomethyl (Cys), variable modifications: Met oxidation, pyroGlu formation from peptide N-terminal Gln residues and acetylation of protein N-termini, allowing maximum 2 variable modifications/peptide; instrument: ion trap. First the Swissprot database (downloaded on 4/16/2015, containing 548208 protein sequences) was searched, then the contaminants identified were appended to the Drosophila melanogaster entries of the Uniprot database concatenated with a randomized sequence for each entry (downloaded on 4/16/2015, containing 42,363 protein sequences). Acceptance criteria: score: 22 and 15; E-values: 0.01 and 0.05 for protein and peptide identifications, respectively (resulting in FDR<0.5% for all samples). All proteins identified in the Vps8-HA immunoprecipitate are listed in *Supplementary file 2*.

## LysoTracker staining and dextran uptake

For LysoTracker Red staining, late L3 stage larval garland nephrocytes were dissected in cold Shields and Sang M3 medium (Sigma Aldrich), and then incubated in medium containing LysoTracker Red (1:1000, Thermo Fisher) for 5 min at room temperature (RT). Samples were rinsed 3 times and photographed immediately. For dextran uptake assay, late L3 stage larval garland cells were dissected in ice cold M3 medium, and then incubated in medium supplemented with Alexa Fluor 568 conjugated dextran (1 mg/ml, fixable, 10000 Da, Molecular Probes) for 5 min at RT, rinsed 3 times and fixed with 4% formaldehyde (FA) in PBS (30 min at RT). Nuclei were counterstained with DAPI in both cases. Pictures were taken on a microscope (AxioImager.Z1; Carl Zeiss) equipped with a grid confocal unit (ApoTome1; Carl Zeiss), using 40×, 0.75 NA (air), and 63×, 1.4 NA (oil) objectives in Lysotracker and dextran experiments, respectively, a CCD camera (AxioCam MRm; Carl Zeiss), and AxioVision software (Carl Zeiss).

## Larval hemocyte preparations

To quantify circulating hemocytes, L3 wandering larvae were dissected on 12-spot microscope slides (Hendley-Essex, Essex, UK) in Drosophila Ringer's solution containing 0.01% 1-phenyl-2-thiourea (Sigma Aldrich). Hemocytes were left to adhere for 45 min, and then fixed with acetone for 6 min.

Samples were mounted with PBS-glycerol and were analyzed with a Zeiss Axioskope 2 MOT microscope (Carl Zeiss) using phase contrast mode. For phagocytosis experiments, L3 wandering larvae were washed in Drosophila Ringer's solution, and placed on a dry paper towel. Larvae were injected with FITC-labeled E. coli using a glass capillary. After 1 hr, larvae were dissected on 12-spot microscope slides in Drosophila Ringer's solution containing 0.01% 1-phenyl-2-thiourea. Hemocytes were left to adhere for 45 min, then lysosomes were stained with 0.1 μM LysoTracker Red DND-99 solution (Thermo Fisher) for 5 min. Surface-bound bacteria were quenched with 0.4% trypan blue solution, and samples were analyzed on an Olympus FLUOVIEW FV1000 confocal laser scanning microscope (Olympus, Tokyo, Japan).

## Immunohistochemistry

Wandering larvae were dissected in cold PBS and fixed with 4% formaldehyde in PBS for 45 min at RT. Samples were extensively washed, and then incubated in 0.1 (v/v) Triton X-100 in PBS (PBTX, 30 min, RT), followed by blocking solution (5.0% (v/v) FCS in PBTX). Samples were then incubated in the blocking solution completed with primary antibodies (overnight, 4°C). Samples were rinsed 3×, washed in PBTX (3 × 10 min at RT), and incubated in blocking solution (30 min at RT). Samples were then incubated with the proper secondary antibodies diluted in blocking solution for 3 hr at RT. Washing steps were repeated, nuclei were stained with DAPI and samples were mounted in PBS-glycerol. LBPA staining was performed as described (*Kim et al., 2010*). The following antibodies were used: monoclonal rat anti-HA 1:80 (Roche), rat anti-Rbsn-5 (1:1000, [*Tanaka and Nakamura, 2008*]), rabbit anti-Rbsn-5 (1:1000, [*Tanaka and Nakamura, 2008*]), rabbit anti-Rab7 (1:1000, [*Tanaka and Nakamura, 2008*]), rabbit anti-Rab5 (1:100, Abcam), mouse anti-LBPA (1:500, Echelon, Salt Lake City, UT), rabbit anti-myc (1:80, EMD Millipore), rabbit anti-Vps16a (1:50, [*Akbar et al., 2009*]), mouse anti-Notch extracellular domain (1:100, DSHB, C458.2H), mouse anti-Pericardin (1:100, DSHB, EC11), rabbit anti-GFP (1:1000, Abcam, Cambridge, UK), rabbit anti-Cathepsin L (1:100, Abcam), rhodamine-phalloidin (0.5 μg/mL). Secondary antibodies were: Alexa Fluor 568 goat anti-Mouse, Alexa Fluor 568 goat anti-Rabbit, Alexa Fluor 568 goat anti-Rat, Alexa Fluor 488 goat anti-Mouse, Alexa Fluor goat anti-Rabbit (all 1:1000, all Invitrogen - Thermo Fisher). Anti-Rbsn-5 and anti-Rab7 were gifts from A. Nakamura, RIKEN Center for Developmental Biology, Japan. Images were obtained on a microscope (AxioImager.Z1; Carl Zeiss) equipped with a grid confocal unit (ApoTome1; Carl Zeiss), using Plan-Neofluar 20×, 0.5 NA (air), 40×, 0.75 NA (air), and 63×, 1.4 NA (oil) objectives, a CCD camera (AxioCam MRm; Carl Zeiss), and AxioVision software (Carl Zeiss). In order to enhance focus depths on high-magnification garland nephrocyte images, images from 3–5 consecutive focal planes (section thickness 0.24 μm) were projected onto one single image. Microscope and imaging settings were identical for experiments of the same kind. Primary images were processed in AxioVision and Photoshop CS4 (Adobe, San Jose, CA) to produce final figures. Note that Alexa Fluor 568 and LysoTracker channels are pseudocolored magenta on most images.

For immunostaining of the circulating hemocytes, the following antibodies were used: anti-NimrodC1 (plasmatocyte specific mouse monoclonal, undiluted) (*Kurucz et al., 2007a*), anti-Atilla (lamellocyte specific mouse monoclonal, undiluted) (*Kurucz et al., 2007b*) and 12F6 (crystal cell specific mouse monoclonal, undiluted) (a kind gift from Prof. Tina Trenczek, University of Giessen). Hemocytes were isolated in Drosophila Ringer's solution containing 0.01% n-phenylthiourea, and were incubated for 45 min at RT. After the incubation, the samples were fixed in acetone for 6 min. This was followed by blocking with 0.1% BSA in 1xPBS for 20 min. The hemocytes were then incubated with the primary antibodies at RT for 1 hr. This was followed by washes with 1xPBS for 5 min, repeated three times. The secondary antibody mixture contained the anti-mouse CF568 antibody diluted 1:1000 and DAPI (Sigma Aldrich) in 0.5 ug/mL concentration in 0.1% BSA-containing PBS. The hemocytes were incubated with the secondary antibody mixture for 1 hr at RT, followed by 3 washes with PBS for 5 min. The slides were mounted with FluoromountG mounting medium (Southern Biotech, Birmingham, AL), and covered by Menzel-Glaser coverslips. Microscopic analysis of the samples was carried out with a Zeiss Axioskop2-MOT epifluorescent microscope, and images were acquired with the AxioVision software (Carl Zeiss).

All stainings have been carried out at least twice (technical replicate), with similar results.

## Electron microscopy

Dissected proventriculi with garland nephrocytes attached were fixed in 3.2% paraformaldehyde, 0.5% glutaraldehyde, 1% sucrose, and 0.028% $CaCl_2$ in 0.1 N sodium cacodylate, pH 7.4 for overnight at 4°C, postfixed in 0.5% osmium tetroxide for 1 hr and in half-saturated aqueous uranyl acetate for 30 min, dehydrated in a graded series of ethanol and embedded into Durcupan (Fluka - Sigma Aldrich) according to the manufacturer's recommendations. 70-nm sections were stained in Reynold's lead citrate and viewed on a transmission electron microscope (JEM-1011; JEOL, Tokyo, Japan) equipped with a digital camera (Morada; Olympus) using iTEM software (Olympus).

## Image analysis and statistics

For colocalization experiments, unmodified raw images were imported in ImageJ software (National Institutes of Health, Bethesda, MD), and the coloc2 plugin was used to calculate Pearson's coefficients (1=perfect colocalization, 0=no/incidental colocalization, −1=mutually exclusive localization) and the Manders coefficients plugin was used to generate scatterplots. For endosome distribution analyses, cropped images (indicated by white rectangles on composite images) were imported in ImageJ, channels were split, and plot profiles were generated using identical settings in each case. Garland cells of at least three animals per genotype were evaluated with similar results, and one representative plot is shown.

For endosome size measurements in fluorescent images, images were imported in ImageJ and the size of endosomes were measured manually. Endosome sizes are profile areas measured in square micron. During endosome size measurements, only Rbsn-5 positive endosomes without Rab7 signal were considered as Rbsn-5 positive early endosomes. Rab7 and Rbsn-5 double positive vesicles were considered as Rab7 positive late endosomes. In the case of the E. coli uptake assay, intact bacteria per cell were counted manually. For endosome size measurements in the ultrastructural analyses, all endosomes with electron lucent lumen and without a clathrin coat were measured manually using iTEM software (Olympus). All visible endosomes were measured in all cells present in the examined section to avoid the distortion of data.

In nephrocyte experiments, 3–5 cells per animal were selected for quantification. Only cells that were optically sectioned in the middle part were included in the analysis to avoid distortion of data or undersampling. All phenotypes shown were robust with no obvious variability between cells or animals of the same genotype, and no data have been purposely excluded from the analysis.

Data were imported into SPSS Statistics 17 (IBM, Armonk, NY) and tested for normality of data distribution. Since in all experiments there were groups with non-normally distributed data, we used nonparametric Kruskal-Wallis test for multiple comparisons. Simple dot plot and box-and-whisker plot charts were generated using the same software. In box-and-whisker plots, bars show the data lying between the upper and lower quartiles; the median is indicated as a horizontal line within the box, and whisker plots show the smallest and largest observations. Outliers are not shown. In simple dot plots, the red lines represent mean values. A p-value of 0.01 (or less) was considered to be significant. Accordingly, ** means $p<0.01$ and *** means $p<0.001$.

Please see *Supplementary file 4* for more details of statistical analyses.

## Acknowledgements

We would like to thank Sarolta Pálfia and Gina Puska for technical assistance, and our colleagues listed in the Methods section for providing reagents. This work was supported by the Wellcome Trust (087518/Z/08/Z to GJ), Hungarian Academy of Sciences (Momentum LP-2014/2 to GJ, and IF-27/2012 to KFM), and Hungarian National Scientific Fund (NK101730 to IA, PD112632 to KH, and K83509 to GJ).

## Additional information

### Funding

| Funder | Grant reference number | Author |
|---|---|---|
| Wellcome Trust | 087518/Z/08/Z | Gábor Juhász |

| | | |
|---|---|---|
| Hungarian Academy of Sciences | Momentum LP-2014/2 | Gábor Juhász |
| Hungarian National Scientific Fund | K83509 | Gábor Juhász |
| Hungarian Academy of Sciences | IF-27/2012 | Katalin F Medzihradszky |
| Hungarian National Scientific Fund | NK101730 | István Andó |
| Hungarian National Scientific Fund | PD112632 | Krisztina Hegedűs |

The funders had no role in study design, data collection and interpretation, or the decision to submit the work for publication.

## Author contributions

PL, Conception and design, Acquisition of data, Analysis and interpretation of data, Drafting or revising the article; ZL, ÁV, TM, ZD, KH, Acquisition of data, Analysis and interpretation of data, Drafting or revising the article; ZS-V, ML, Acquisition of data, Drafting or revising the article; PB, GC, Acquisition of data, Analysis and interpretation of data; IA, KFM, Analysis and interpretation of data, Drafting or revising the article; ST, GJ, Conception and design, Analysis and interpretation of data, Drafting or revising the article

## Author ORCIDs

Péter Lőrincz, http://orcid.org/0000-0001-7374-667X
Zsolt Lakatos, http://orcid.org/0000-0003-1900-3167
Tamás Maruzs, http://orcid.org/0000-0001-8142-3221
Zsófia Simon-Vecsei, http://orcid.org/0000-0001-7909-4895
Gábor Juhász, http://orcid.org/0000-0001-8548-8874

# Additional files

### Supplementary files

• Supplementary file 1. Nucleotide sequence of the wild type vps8/CG10144 gene (from the translational start codon to the stop codon) and the deletions present in the vps8[1] allele.

• Supplementary file 2. Detailed Vps8-HA proteomic data. The table contains all peptides/proteins identified in the precipitate from Vps8-HA animals, which were absent in controls. Acc#: UNIPROT accession number. Num Unique: number of unique peptides identified for the listed protein.% Cov: The percentage indicates the sequence coverage by the identified peptides. MW: molecular weight in Daltons. Both% Cov and MW were calculated from the genomic sequence listed in the database that may differ from the mature protein. DROME: *Drosophila melanogaster*.

• Supplementary file 3. Genotype of animals used in this study.

• Supplementary file 4. Additional table showing statistical tests, N values and p-values.

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
