## [Decision Letter]

Thank you for submitting your article "MiniCORVET is a Vps8-containing hemocyte- and nephrocyte-specific early endosomal tether in *Drosophila*" for consideration by *eLife*. Your article has been reviewed by three peer reviewers, and the evaluation has been overseen by Randy Schekman as the Senior and Reviewing Editor. Two reviewers, Alexey Merz and Helmut Krämer, have agreed to be identified. The reviewers have discussed the reviews with one another and the Reviewing Editor has drafted this decision to help you prepare a revised submission.

Summary:

Lorincz and co-workers report thorough and elegant genetic studies on *Drosophila* of Vps8, a substituent of CORVET, one of multiple Vps-C multi-subunit tethering complexes that support membrane docking and fusion within the endolysosomal system. Perhaps as expected from previous work in other systems, they find that Vps8 is needed for the fusion and steady-state maintenance of an early, Rabenosyn-5 positive compartment. A second Vps-C compartment, HOPS, is needed to control docking and steady-state maintenance of a late-endosomal, Rab7-positive compartment. This is also expected. However, Lorincz et al. reveal some surprises as well. First, Vps8 exhibits a strongly tissue-specific localization to nephrocytes and hemocytes, but is doesn't seem to be needed for endosomal function in wing imaginal discs. Second, Vps8 deletion causes expansion of the hemocyte lineage and tumorigenesis. Third, Vps11, long thought to be a core component required for all Vps-C complexes, is dispensable for early endosome function in the hemocyte but still needed (along with HOPS at the late endosome. Interaction studies are consistent with this interpretation and reveal a "mini-CORVET" complex seemingly containing four rather than the expected six subunits.

The reviewers agree on the importance and rigor of the study, medium to high novelty and the surprises of no vps11 and cell-specific expression.

Essential revisions:

1) The first question relates to the specificity of vps8 expression and function. I am not sure specific loss of CORVET function in hemocytes and nephrocytes in larvae alone would cause pharate adult lethality, as the authors observe for the vps8 CRISPR mutant. I am very surprised that the endogenously tagged HA variant is not explained either in Results or in the Materials and methods. Only presenting primer sequences is absolutely insufficient – I did not want to make the effort to BLAST the primers myself to find out how big the region may be – but any PCR amplified region is highly likely to represent only some genomic regulatory elements. How big is the region? Did they use CRISPR or BAC recombineering, and where is the transgene integrated in the genome? It actually seems like the construct does not fully rescue the defects of the vps8/Df in Figure 4 and Figure 8. Next, there is no data where else Vps8-HA is expressed in the larva or in the pupa. Why do they die as late pupae? Do I understand correctly that adult escapers are actually perfectly normal? How about the nervous system?

Minimally the authors should show and discuss: first, exactly how the genomically tagged version was made and where it is expressed in larvae, pupae and adults. Second, a more detailed and thoughtful presentation of rescue is warranted.

2) There are (presentational?) problems with the vps8 mutant. Figure 3 shows that vps8 transcript is still present in the RTPCR result, so it is not convincing that vps8 mutant they generated is really a mutant. There seems to be a decrease in the level of vps8 transcript in vps8/Df compared to the control, but this would even be consistent with just a loss of one copy of vps8 in the deficiency line. How about the homozygous vps8 mutant? Why is there any RTPCR product? Are there any differences between vps8 homozygous CRISPR allele and vps8 over Deficiency? I acknowledge that the Vps8-HA apparently rescues semilethality, making this concern hopefully only one of presentation of full data. Is a homozygous vps8 mutant rescued equally to the transheterozygote?

3) Figure 6: Why does the western blot show no anti-HA band in the Input column when Vps8-HA is present (+ column).

4) *Drosophila* pigment cells are another cell type that relies on HOPS or a related complex for the biogenesis of a specialized lysosome-related organelle. So it was disappointing that the possible function for Vps8 in pigment granule formation was not assessed, given that it requires only a simple cross in a white+ background. An examination of pigment granule phenotypes is warranted, particularly given the long and rich history of Vps-C studies in that context. Probably the first SM allele ever isolated, even before sec1-1, was fly carnation/Vps33A. And, now that I mention it, that paper should be cited in the present manuscript: J.T. Patterson (1932) Genetics 17:38-59.

5) The observation of hemocyte and nephrocyte-specificity is not well documented – the genomic (?) HA construct may not reflect the normal expression pattern, is insufficiently explained, and expression in other than the analyzed cell types simply not shown. There is also a potential issue with the mutant. If all data are correct and the reagents good, then this should be easy to revise for the authors.

---

## [Author Response]

*Essential revisions:*

*1) The first question relates to the specificity of vps8 expression and function. I am not sure specific loss of CORVET function in hemocytes and nephrocytes in larvae alone would cause pharate adult lethality, as the authors observe for the vps8 CRISPR mutant. I am very surprised that the endogenously tagged HA variant is not explained either in Results or in the Materials and methods. Only presenting primer sequences is absolutely insufficient – I did not want to make the effort to BLAST the primers myself to find out how big the region may be – but any PCR amplified region is highly likely to represent only some genomic regulatory elements. How big is the region? Did they use CRISPR or BAC recombineering, and where is the transgene integrated in the genome? It actually seems like the construct does not fully rescue the defects of the vps8/Df in Figure 4 and Figure 8. Next, there is no data where else Vps8-HA is expressed in the larva or in the pupa. Why do they die as late pupae? Do I understand correctly that adult escapers are actually perfectly normal? How about the nervous system?*

Minimally the authors should show and discuss: first, exactly how the genomically tagged version was made and where it is expressed in larvae, pupae and adults. Second, a more detailed and thoughtful presentation of rescue is warranted.

Thank you for the suggestion to include more data on our Vps8-HA reporter/rescue transgene. We have included it in Figure 1—figure supplement 1, and a detailed description of this construct in its legend as follows:

“This construct contains 824 base pairs upstream of the Vps8 transcription start site (including part of the first exon of the neighboring gene sfl), the entire Vps8 transcription unit including exons and introns, but lacks the stop codon and 3' UTR of Vps8. […] We show the X chromosomal insertion throughout the paper, because it was easier to carry out genetic rescue experiments with that (note that the Vps8 gene is located on the 3rd chromosome).”

In the revised version of our manuscript, we have included additional images showing that Vps8-HA (which is highly expressed in nephrocytes and hemocytes, as shown in Figure 1) shows much lower levels of expression in other tissues in the larva (Figure 1—figure supplement 1). Its expression pattern in various adult tissues and pupal wings is similar to larvae (Figure 1—figure supplement 2). This is perfectly in line with high-throughput expression data available on Flybase, the *Drosophila* genome homepage: both the FlyAtlas and modENCODE datasets showed that the expression of Vps8 is low in most tissues approximately 20-200-fold lower that of the common autophagy marker Atg8a). Note that hemocytes and nephrocytes were not analyzed in these high-throughput studies, probably because the collection of enough samples from these cells was not feasible.

We have also included more details for genetic rescue experiments. While Vps8-HA does not revert the Rab7 vesicle/α vacuole size defect of Vps8 mutants completely to wild type (as shown in Figure 5 and Figure 8, respectively), the rescued animals were indistinguishable from wild type in most tests (including vps8 mRNA levels and viability, please see Figure 3 and Figure 3—figure supplement 1). We hypothesize that the presence of the C-terminal 9xHA tag might minimally interfere with the generation of the unusually large Rab7-positive alpha vacuoles in nephrocytes.

The most likely explanation for the lethality of our Vps8 mutant is, at least in part, spontaneous immune induction. Immune induction is clearly indicated by the disorganization of the sessile hemocyte compartment and en masse appearance of specialized blood cells (lamellocytes and crystal cells, please see Markus et al., 2009, PNAS 106:4805-4809) in mutant larvae. New images showing these phenotypes have been added, please see Figure 3—figure supplement 1. Abnormal immune induction and hemocyte proliferation has previously been shown to cause lethality due to hematopoetic compartment-specific overexpression of PVF2, a PDGF/VEGF-like growth factor (Munier et al. 2002, EMBO reports vol. 3 no.12 pp1195–1200). These two references have also been added to the revised manuscript. Vps8-HA expression is very low in the nervous system (Figure 1—figure supplement 1, and Figure 1—figure supplement 2). Since the overall brain morphology of dissected vps8 mutant larvae and adults appears similar to wild type, and we could not find any obvious alterations in various vps8 mutant neuron types compared to wild type animals in ultrastructural analyses (our unpublished observations), we hypothesize that Vps8 might not play a major role in neurons.

2) There are (presentational?) problems with the vps8 mutant. Figure 3 shows that vps8 transcript is still present in the RTPCR result, so it is not convincing that vps8 mutant they generated is really a mutant. There seems to be a decrease in the level of vps8 transcript in vps8/Df compared to the control, but this would even be consistent with just a loss of one copy of vps8 in the deficiency line. How about the homozygous vps8 mutant? Why is there any RTPCR product? Are there any differences between vps8 homozygous CRISPR allele and vps8 over Deficiency? I acknowledge that the Vps8-HA apparently rescues semilethality, making this concern hopefully only one of presentation of full data. Is a homozygous vps8 mutant rescued equally to the transheterozygote?

Thank you for the suggestion to clarify these. The CRISPR mutagenesis and the resulting mutations in vps8^1^ are now shown in Figure 3. The first mutation causes a 4 base pair deletion, and this frameshift immediately produces a premature stop codon after amino acid 39 (note that Vps8 protein is composed of 1,229 amino acids). We thus think that vps8^1^ represents a null allele. In line with this, the phenotypes of homozygous vps8^1^ mutants are similar to vps8^1^/Df animals, and both are rescued equally by Vps8-HA. Please see Figure 3 for viability data, Figure 4—figure supplement 1 for silver and dextran uptake, Figure 5—figure supplement 1 for Rab7 vesicles in homozygous mutants and rescue animals. Quantification of dextran uptake, Rab7 endosome size and Garland cell size in homozygous and Vps8/Df animals as well as in rescued animals is shown in Figure 4—figure supplement 1, and in Figure 5 respectively.

We have carried our quantitative real-time PCR experiments to properly assess vps8 transcript levels in the various genotypes. Please note that vps8^1^ animals carry a frameshift mutation, which likely does not affect the transcription of this gene but causes a premature stop codon as shown now in Figure 3 as well as in [Supplementary-material SD1-data]. As expected, qPCRs using primers overlapping with the frameshift mutation do not detect wild type vps8 transcript in homozygous and vps8/Df animals, and it is restored to wild type level in the rescued genotypes (Figure 3). We also carried out qPCR analysis using frameshift-independent primers, which show approximately 25% transcript level in homozygous and vps8/Df animals relative to wild type (Figure 3—figure supplement 1). This is similar to what we have shown by RT-PCR in the original version of our manuscript (note that this is replaced with qPCR data in the revised version). In summary, the frameshift mutation does not prevent the transcription of this locus, but it likely affects mRNA stability.

3) Figure 6: Why does the western blot show no anti-HA band in the Input column when Vps8-HA is present (+ column).

Thank you for this suggestion. We have carried out this western blot again using the original input lysates and longer exposure time, and replaced the image with one where the anti-HA band is visible (please see Figure 6).

4) Drosophila pigment cells are another cell type that relies on HOPS or a related complex for the biogenesis of a specialized lysosome-related organelle. So it was disappointing that the possible function for Vps8 in pigment granule formation was not assessed, given that it requires only a simple cross in a white+ background. An examination of pigment granule phenotypes is warranted, particularly given the long and rich history of Vps-C studies in that context. Probably the first SM allele ever isolated, even before sec1-1, was fly carnation/Vps33A. And, now that I mention it, that paper should be cited in the present manuscript: J.T. Patterson (1932) Genetics 17:38-59.

Thank you for the suggestion to analyze eye pigment granule formation. We have replaced Figure 3 (with the eyes shown enlarged in C'), showing that the eye color of Vps8 mutant males and females is indistinguishable from that of wild type flies.

The first report of carnation mutant flies (Patterson, J.T. (1932). Lethal mutations and deficiencies produced in the X-chromosome of *Drosophila melanogaster* by X-radiation. Am. Nat. 66: 193-206) is now cited. Please note that the Genetics paper mentioned by the Reviewer describes two translocations that, to our knowledge, do not affect the carnation gene.

*5) The observation of hemocyte and nephrocyte-specificity is not well documented – the genomic (?) HA construct may not reflect the normal expression pattern, is insufficiently explained, and expression in other than the analyzed cell types simply not shown. There is also a potential issue with the mutant. If all data are correct and the reagents good, then this should be easy to revise for the authors.*

We have included more data on Vps8-HA expression driven by a genomic promoter region (please see response to Essential revision 1) and on our vps8 mutant (please see response to Essential revision 2), as requested. We suggest that *Drosophila* Vps8 has the most important function in hemocytes and nephrocytes because:

A) we can detect striking hemocyte and nephrocyte defects in our vps8 mutant flies;

B) these are the cell types where our genomic Vps8-HA reporter shows high expression;

C) the mutant phenotypes, including lethality and melanotic tumors, are rescued by our Vps8-HA transgene, indicating that these are caused by the loss of vps8 function and that the Vps8-HA protein is functional.

Nevertheless, since Vps8 expression can also be detected in other tissues in low levels, we refrain from stating hemocyte- and nephrocyte-specificity in the revised manuscript.